# Offline Reinforcement Learning via Inverse Optimization

**Ioannis Dimanidis**                                                  *ioanndima@gmail.com*
*Delft Center for Systems and Control*
*Delft University of Technology*
*Netherlands*

**Tolga Ok**                                                           *tok@tudelft.nl*
*Delft Center for Systems and Control*
*Delft University of Technology*
*Netherlands*

**Peyman Mohajerin Esfahani**                          *P.MohajerinEsfahani@utoronto.ca*
*Industrial & Mechanical Engineering*
*University of Toronto*
*Canada*

**Reviewed on OpenReview:** *https://openreview.net/forum?id=lSj2gXoeCy*

## Abstract

Inspired by the recent successes of Inverse Optimization (IO) across various application domains, we propose a novel offline Reinforcement Learning (ORL) algorithm for continuous state and action spaces, leveraging the convex loss function called "sub-optimality loss" from the IO literature. To mitigate the distribution shift commonly observed in ORL problems, we further employ a robust and non-causal Model Predictive Control (MPC) expert steering a nominal model of the dynamics using in-hindsight information stemming from the model mismatch. Unlike the existing literature, our robust MPC expert enjoys an exact and tractable convex reformulation. In the second part of this study, we show that the IO hypothesis class, trained by the proposed convex loss function, enjoys ample expressiveness and reliably recovers teacher behavior in MuJoCo benchmarks. The method achieves competitive results compared to widely-used baselines in sample-constrained settings, despite using orders of magnitude fewer parameters. To facilitate the reproducibility of our results, we provide an open-source package implementing the proposed algorithms and the experiments. The code is available at `https://github.com/TolgaOk/offlineRLviaIO`.

## 1 Introduction

In dynamic environments where real-world interactions are impractical, there is often the need to work with datasets of previously collected interactions. Decision-making in these contexts typically follows one of two paradigms. (i) Imitation learning (IL), a subclass of the Supervised Learning (SL) paradigm, in which the aim is to imitate a given expert's decisions (i.e., labels in SL terms) and (ii) offline Reinforcement Learning (RL), where the aim is to learn a policy that improves upon the performance observed within the dataset. SL in general, and IL in particular, has proven to be successful in a wide range of applications (Hussein et al., 2017), while offline RL is known to be a notoriously hard task (both computationally and statistically) (Bertsekas, 2021). One of the primary challenges in offline RL is the mismatch between the dataset and the policy distributions. Hence, naively applying existing online RL algorithms combined with high-capacity Q function approximation leads to optimistic and potentially biased value functions, which, in turn, leads to poorly performing and unstable policies that do not generalize in the online evaluation.

To combat these issues, in this work, we approach the offline RL problem in two steps: (i) by utilizing a non-causal expert, we perform an "action improvement" step over the dataset; and (ii) using the improved actions, we fit a Q-function using a novel "sub-optimality loss" to obtain an efficient and causal policy that generalizes over online evaluations. Specifically, in the first step, by leveraging a nominal model and in-hindsight model mismatch information, unknown at runtime, we introduce an expert in the form of a non-causal Model Predictive Control (MPC). While we assume access to a nominal model, the true dynamics are influenced by unknown disturbances and model mismatches. A traditional MPC approach in this setting would either ignore these disturbances—leading to significantly degraded performance—or require significant human effort to manually model the residuals and incorporate them into the control loop. By contrast, our non-causal teacher/causal student setup streamlines this process: the non-causal expert utilizes in-hindsight data to determine the optimal response to disturbances, which the student then distills into a causal policy. This enables the agent to learn directly from the data distribution without the need for explicit disturbance modeling or heavy human supervision. To realize the non-causal expert, we propose to replace the Bellman residual loss with the "sub-optimality loss" drawn from the Inverse Optimization (IO) literature that fits the optimal Q function given the improved dataset. The proposed optimization problem enjoys the convexity of the loss function, yielding an efficient and causal policy that can generalize over unseen states. Before proceeding with further details regarding our proposed approach and the related literature, we introduce some notations.

**Notation:** The dimension of a variable $x$ is denoted by $n_x$. We denote with $N$ the MPC horizon and with $T$ the size of a dataset. With bold, we denote the stacking of variables, i.e., $\mathbf{x}_{1:N} = (x_1, x_2, \ldots, x_N)$, unless noted otherwise. When no exact range is given in the subscript, the default length of a bold variable is $N$ (i.e., $\mathbf{x} = \mathbf{x}_{1:N}$). We denote by $\langle \cdot, \cdot \rangle$ an inner product with the respective norm $\|x\|^2 = \langle x, x \rangle$. For any $A \succ 0$, we define $\|x\|_A^2 = \langle x, Ax \rangle$. With $\otimes$, we denote the Kronecker product. As the letter "Q" will be used to indicate both matrices and Q-functions, we denote with $Q$ the former and with $\mathrm{Q}(s, u)$ the latter, although it should usually be clear from the context. The operators $\mathrm{diag}(\cdot)$ and $\mathrm{blkdiag}(\cdot)$ construct a square or block matrix, respectively. Finally, with MPC-$N$, we refer to policies stemming from the minimization of an $N$-stage cost that predicts the future behavior of the system using some model.

## 1.1 Problem statement and Contributions

We consider the constrained control of discrete-time dynamical systems with unknown dynamics $f$, where we have access to a deterministic nominal model $f_0$, and an offline trajectory of state-action pairs $\mathcal{D}_T = \{\hat{x}_t, \hat{u}_t\}_{t=0}^T$ collected under some behavior policy applied to $f$. The control $u$ is constrained to belong to some $\mathcal{U}(x)$, which is also assumed to be known, and there is an $N$-stage control objective defined through the known stage- and terminal-cost functions $c(x, u)$ and $c_f(x)$. We aim to learn a causal[1] stationary parameterized policy $\pi_\theta$ that distills a non-causal MPC expert on $f_0$ with access to the full future model-mismatch sequence inferred from $\mathcal{D}_T$ and $f_0$. Concretely, our proposed RL scheme comprises two key steps:

- *Non-causal action improvement:* From $\mathcal{D}_T$ and $f_0$ infer the mismatch trajectory (e.g., $\hat{w}_{t+1} = \hat{x}_{t+1} - f_0(\hat{x}_t, \hat{u}_t)$). Feed the full future mismatch sequence into an $N$-stage receding-horizon non-causal MPC defined on $f_0$ to produce an expert control sequence $\{\hat{u}_t^{\mathrm{ex}}\}_{t=1}^T$. A robust variant replaces the known mismatch with a worst-case element from a specified uncertainty set.

- *Imitation learning/policy distillation:* Fit a causal policy $\pi_\theta$ to the new improved state-action dataset $\{\hat{x}_t, \hat{u}_t^{\mathrm{ex}}\}_{t=1}^T$ via a tractable convex IO objective (sub-optimality loss) yielding a computationally cheap policy for deployment.

The above procedure is documented in detail in Algorithm 1. Building on this setting, we summarize our contributions as follows:

(i) **Two-step offline RL via IO-based policy distillation:** We use the Inverse Optimization framework to bridge offline RL with Imitation Learning, distilling the non-causal MPC expert from the above

---

[1]A policy $\pi$ is deemed to be causal iff it depends only on past and present data, i.e., $u_t = \pi(s_\tau | \tau \le t)$.

two-step process into a causal policy. This approach circumvents the need for explicit disturbance models by learning the environment's nuances directly from trajectory data. The resulting formulation is computationally attractive due to the convexity of the training landscape and results in policies that are efficient to evaluate, while also opening the door for tools from online convex optimization to be readily used for the control tasks considered here.

(ii) **Tractable robustification of the MPC expert:** For the case of linear dynamics, quadratic stage/terminal costs, and polytopic constraints, we derive an exact convex reformulation of a robust non-causal MPC expert that optimizes against worst-case model-mismatch trajectories within a prescribed uncertainty set. This allows us to incorporate adversarial robustness without introducing conservatism or sacrificing tractability. From the empirical analysis of Appendix B and Section 4.1, we show that the robustification helps combat the distribution shift from the training to the test phase and the mismatch between the nominal model and the true dynamics.

(iii) **Empirical validation of IO expressiveness:** Through experiments on nonlinear control problems and MuJoCo benchmarks, we provide evidence that the IO hypothesis class is expressive enough for high-quality policy distillation in imitation learning. In particular, our method achieves state-of-the-art performance in low-data regimes while using orders-of-magnitude fewer parameters than neural-network-based baselines.

## 1.2 Related works

**Offline Reinforcement Learning:** To prevent the value function from exploiting any dataset bias, offline RL approaches typically attempt to enforce pessimistic policy learning (Rashidinejad et al., 2021); this can be achieved by constraining the policy learning within the region supported by the dataset (Fujimoto et al., 2018) or by penalizing the value function for the state-action pairs outside the dataset (Kumar et al., 2019; Wu et al., 2019; Kostrikov et al., 2021; Kumar et al., 2020). Model-based approaches employ similar ideas but instead try to exploit the model information to learn a less conservative value function. For instance, COMBO (Yu et al., 2021) approximates the true model dynamics and utilizes both simulated and dataset samples to learn a conservative value estimation by penalizing out-of-support state-action pairs obtained by running the simulated model. On the other hand, our proposal uses a nominal model to improve the actions of the state-action pairs present in the dataset; finally, in contrast to the aforementioned works, our work is more computationally attractive, as the resulting program for learning the policy is convex.

**Imitation Learning:** The second step of our algorithm, where we employ Inverse Optimization to fit a policy on the improved state-action pairs, is analogous to IL. Similar to our dataset improvement scheme, several other IL algorithms employ augmentation strategies to further improve policy learning. For example, BAIL (Chen et al., 2020) first estimates the Monte Carlo returns of each state-action pair in the dataset, an infinite horizon and discounted extension of our objective function, and employs a neural network-based estimate to fit the returns. Based on this estimate, BAIL selects only the highest-valued state-action pairs and learns a policy via IL. On the other hand, our approach makes use of the entire dataset, improving actions through our robust MPC formulation, and utilizes a convex "sub-optimality loss" to perform the IL step. The decision to employ a relatively small model together with a convex loss function (in the parameter space) is justified by our empirical studies (see Section 4) and aligns with similar findings reported by Emmons et al. (2021), particularly in limited data regimes.

**Terminal value function approximation:** Since MPC projects its internal model into the future, it can also act as an approximation to the Bellman equation. This observation is exploited by (Zhong et al., 2013) to effectively increase the planning horizon by constructing approximate terminal Value Functions (VF) from MPC simulation data. Using the same principle, (Lowrey et al., 2018) showcased an algorithm that promotes exploration and, therefore, accelerates VF learning. Finally, (Bhardwaj et al., 2020) propose a blended approach that combines elements from model-free and model-based methods to reduce model bias. Similarly, our work can be viewed as a specific instance of VF approximation, where learning the Q-function reduces the horizon to a single step. Additionally, in contrast to the papers mentioned above, our approach is computationally tractable.

**Trajectory Augmentation:** A large body of work has studied the augmentation of offline datasets to mitigate distributional shift and synthetically generate high-return trajectories. Among recent approaches, Generative Trajectory Augmentation (GTA) (Lee et al., 2024) employs a trajectory-level conditional diffusion model to enrich the offline RL dataset toward high-return regions, while remaining consistent with observed data. Similarly, Diffusion-based Trajectory Stitching (DiffStitch) (Li et al., 2024) synthesizes bridging sub-trajectories that connect low- and high-return trajectories, effectively *stitching* them together to form an expanded dataset. More closely related to our setting are model-based augmentation methods (Wang et al., 2021; Lyu et al., 2022; Zhang et al., 2023), which learn the dynamics and the rollout policy to generate synthetic trajectories, while enforcing conservatism via model agreement or uncertainty-based truncation. By contrast, we do not synthesize new states or transitions; instead, given a nominal model, we *improve* the actions along observed trajectories by solving the robust non-causal MPC problem introduced in Section 3.

## 2 Inverse Optimization for RL

In what follows, we briefly review the existing literature on IO and its potential to learn a control law. We then introduce the first contribution of this study: how in-hindsight information can be exploited to devise an offline RL algorithm.

### 2.1 Inverse Optimization as Supervised Learning

The goal of Inverse Optimization is to learn the behavior of an expert whose actions depend on an external signal. Specifically, for a given $s \in \mathcal{S} \subseteq \mathbb{R}^{n_s}$, the expert's decisions $u^{\mathrm{ex}} \in \mathcal{U}(s) \subseteq \mathbb{R}^{n_u}$ stem from a deterministic policy: $u^{\mathrm{ex}} = \pi^{\mathrm{ex}}(s)$. We wish to approximate $\pi^{\mathrm{ex}}(s)$ with a policy in a similar spirit as in Q-Learning that is defined as:

$$\pi_\theta(s) := \underset{u \in \mathcal{U}(s)}{\arg \min} \, \mathrm{Q}_\theta(s, u),$$

where $\mathrm{Q}_\theta$ is a parameterized function belonging to the hypothesis class $\mathcal{Q}$. Throughout this work, we consider the strongly convex quadratic hypothesis class

$$\mathcal{Q} = \{\mathrm{Q}_\theta(s, u) = \langle u, \theta_{uu} u \rangle + 2 \langle s, \theta_{su} u \rangle : \theta_{uu} \succcurlyeq I_{n_u}\}. \tag{1}$$

To learn the optimal $\theta^\star$, we use the "sub-optimality loss", which was first introduced in (Mohajerin Esfahani et al., 2018):

$$\ell_\theta^{\mathrm{sub}}(s, u^{\mathrm{ex}}) = \mathrm{Q}_\theta(s, u^{\mathrm{ex}}) - \min_{u \in \mathcal{U}(s)} \mathrm{Q}_\theta(s, u). \tag{2}$$

Notice that the mapping $\theta \mapsto \mathrm{Q}_\theta(s, u)$ is linear, and thus, the "sub-optimality loss" (2) is convex in $\theta$ for convex $\mathcal{U}(s)$. Given a dataset $\{(\hat{s}_t, \hat{u}_t^{\mathrm{ex}})\}_{t=1}^T$ of states $\hat{s}_t$ and expert actions $\hat{u}_t^{\mathrm{ex}} = \pi^{\mathrm{ex}}(\hat{s}_t)$, and a polytopic constraint set $\mathcal{U}(s) = \{u : \ G(s) u \leq h(s)\}$, we have that (Akhtar et al., 2021):

$$\min_{\theta \in \Theta} \sum_{t=1}^T \ell_\theta^{\mathrm{sub}}(\hat{s}_t, \hat{u}_t^{\mathrm{ex}}) = \min_{\theta, \boldsymbol{\gamma}_{1:T}, \boldsymbol{\lambda}_{1:T}} \quad \sum_{t=1}^T \mathrm{Q}_\theta(\hat{s}_t, \hat{u}_t^{\mathrm{ex}}) + \tfrac{1}{4}\gamma_t + \langle \hat{h}_t, \lambda_t \rangle \tag{3}$$

$$\text{s.t.} \qquad \theta_{uu} \succcurlyeq I_{n_u}, \quad \lambda_t \geq 0, \qquad t \leq T,$$

$$\begin{bmatrix} \theta_{uu} & \hat{G}_t^\mathsf{T} \lambda_t + 2\theta_{su}^\mathsf{T} \hat{s}_t \\ \star & \gamma_t \end{bmatrix} \succcurlyeq 0, \quad t \leq T,$$

where we use the shorthand $\hat{G}_t = G(\hat{s}_t)$ and $\hat{h}_t = h(\hat{s}_t)$. The convex optimization (3) offers an efficient way to learn the policy $\pi^{\mathrm{ex}}(\cdot)$. It is important to highlight that a key part upon which this program is built is the sequence of the "ground-truth" expert actions $\hat{\mathbf{u}}_{1:T}^{\mathrm{ex}}$. While the actions contained within an offline RL dataset can be regarded as expert actions, we propose to improve them by leveraging the hindsight information of a controlled dynamical system.

## 2.2 Imitating an MPC expert with Inverse Optimization

Given a deterministic nominal model $f_0$, and denoting state and input constraints for each step as $X$ and $U$ respectively, we formulate the deterministic MPC-$N$ problem as follows:

$$V_N^{\mathrm{mpc}}(x) := \min_{\mathbf{u}} \quad \sum_{k=0}^{N-1} c(x_k, u_k) + c_f(x_N) \tag{4}$$
$$\text{s.t.} \quad \mathbf{u} \in \mathcal{U}_N^{\mathrm{mpc}}(x).$$

where $\mathcal{U}_N^{\mathrm{mpc}}(x) := \left\{\mathbf{u} \in \mathbb{R}^{Nn_u} : u_k \in U, x_{k+1} = f_0(x_k, u_k) \in X, k \le N, x_0 = x\right\}$. Thanks to the principle of optimality, we can express the Q-function of (4) as $\mathrm{Q}^{\mathrm{mpc}}(x, u) = c(x, u) + V_{N-1}^{\mathrm{mpc}}(f_0(x, u))$, which is defined over the 1-step constraint set $\mathcal{U}_1^{\mathrm{mpc}}(x) := \{u \in \mathbb{R}^{n_u} : u \in \mathcal{U}, f_0(x, u) \in \mathcal{X}\}$. To approximate $\mathrm{Q}^{\mathrm{mpc}}$ with Inverse Optimization, we solve (3) with $\hat{s}_t = \hat{x}_t$, and $\hat{u}_t^{\mathrm{ex}} = \pi^{\mathrm{mpc}}(\hat{x}_t)$, where

$$\pi^{\mathrm{mpc}}(x) = \operatorname*{arg\,min}_{u \in \mathcal{U}_1^{\mathrm{mpc}}(x)} \mathrm{Q}^{\mathrm{mpc}}(x, u). \tag{5}$$

**Remark 2.1** (Approximating MPC with IO). For the MPC problem (4) to be tractable, a common assumption is that $f_0$ is linear in $x$ and $u$ and the sets $X$ and $U$ are polytopic. In such a setting, the Q-function $\mathrm{Q}^{\mathrm{mpc}}$ is piecewise quadratic where the number of pieces may be exponential in the horizon length $N$. Therefore, approximating $\mathrm{Q}^{\mathrm{mpc}}$ using the quadratic hypothesis class (1) may likely not be exact. Nonetheless, as reported in (Akhtar et al., 2021), such an approximation can work quite well. If there are no constraints, then (4) becomes a finite-horizon LQR problem, whose Q-function is known to be quadratic and positive definite, and as such, we can have an exact approximation within the hypothesis class (1). In this case, the approximate policy becomes $\pi_\theta(s) = -\theta_{uu}^{-1}\theta_{su}^\intercal s$, which implies that we essentially learn an optimal linear control policy.

## 2.3 Exploiting in-hindsight information

This section contains the first contribution of this study, aiming to bridge the gap between IO and offline RL settings. To this end, we consider an extended nominal model with additive disturbances $w \in \mathbb{R}^{n_w}$, i.e., $\tilde{f}_0(x, u, w) = f_0(x, u) + Ew$ where $E^\dagger E = I$. Denoting the $N$-length disturbance trajectory by $\mathbf{w}$, we define the non-causal MPC-$N$ problem via

$$V_N^{\mathrm{nc\text{-}mpc}}(x, \mathbf{w}) := \min_{\mathbf{u}} \quad \sum_{k=0}^{N-1} c(x_k, u_k) + c_f(x_N) \tag{6}$$
$$\text{s.t.} \quad \mathbf{u} \in \mathcal{U}_N^{\mathrm{nc\text{-}mpc}}(x, \mathbf{w}).$$

with $\mathcal{U}_N^{\mathrm{nc\text{-}mpc}}(x, \mathbf{w}) := \{\mathbf{u} \in \mathbb{R}^{Nn_u} : u_k \in U, x_{k+1} = \tilde{f}_0(x_k, u_k, w_{k+1}) \in X, k \le N, x_0 = x\}$. Then, akin to Section 2.2, we can define $\mathrm{Q}^{\mathrm{nc\text{-}mpc}}(x, u, \mathbf{w})$ and $\mathcal{U}_1^{\mathrm{nc\text{-}mpc}}(x, \mathbf{w})$ accordingly, and therefore we obtain the non-causal MPC expert policy

$$\pi^{\mathrm{nc\text{-}mpc}}(x, \mathbf{w}) = \operatorname*{arg\,min}_{u \in \mathcal{U}_1^{\mathrm{nc\text{-}mpc}}(x, \mathbf{w})} \mathrm{Q}^{\mathrm{nc\text{-}mpc}}(x, u, \mathbf{w}) \tag{7}$$

We construct expert actions by leveraging *in-hindsight disturbance trajectories* extracted from data. Given the extended nominal model $\tilde{f}_0$ and a measured transition $(\hat{x}, \hat{u}, \hat{x}_+)$, we define the residual $E\hat{w} = \hat{x}_+ - f_0(\hat{x}, \hat{u})$. By concatenating such residuals across the dataset $\mathcal{D}_T$, we obtain disturbance sequences that can be injected into the non-causal MPC problem (6) to compute expert actions. Because this policy requires future disturbances $\mathbf{w}_{t+1:t+N}$, not available online at time $t$, it is inherently non-causal and can only be used offline.

To make this expert usable in practice, we approximate it causally. We introduce a feature map $\phi$ that summarizes past information and define the augmented state $s_t = \phi(\mathbf{x}_{1:t}, \mathbf{u}_{1:t})$. In general, the design of $\phi$ is a feature-engineering problem and lies outside the scope of this paper; we assume that, given the application at hand, one has access to features that can capture predictive structure in the disturbances; for example, if disturbances evolve linearly, a natural feature choice is the most recent $H$ residuals, i.e., $\phi(\mathbf{x}_{1:t}, \mathbf{u}_{1:t}) = \mathbf{w}_{t-H+1:t}$, with $H$ chosen sufficiently large.

---

**Algorithm 1** Using in-hindsight information for IO

---
1: **Input:**
2:   - Offline trajectory $\mathcal{D}_T = \{(\hat{x}_t, \hat{u}_t)\}_{t=1}^T$
3:   - Extended nominal model $f_0(x, u) + Ew$
4:   - Non-causal expert policy $\pi_N^{\mathrm{nc}}(x, \mathbf{w})$ with horizon $N$
5:   - Feature map $\phi(\cdot, \cdot)$
6: Initialize dataset of training pairs $\mathcal{D}_{\mathrm{ex}} \leftarrow \emptyset$
7: **for** $t = 1$ **to** $T - 1$ **do**
8:     Compute residual mismatch $\hat{w}_{t+1} \leftarrow E^\dagger(\hat{x}_{t+1} - f_0(\hat{x}_t, \hat{u}_t))$
9:     Compute augmented state $\hat{s}_t \leftarrow \phi(\hat{\mathbf{x}}_{1:t}, \hat{\mathbf{u}}_{1:t})$
10:    Let $\tau \leftarrow t - N + 1$
11:    **if** $\tau \geq 1$ **then**
12:        Query expert action $\hat{u}_\tau^{\mathrm{ex}} \leftarrow \pi_N^{\mathrm{nc}}(\hat{x}_\tau, \hat{\mathbf{w}}_{\tau+1:\tau+N})$
13:        Append $(\hat{s}_\tau, \hat{u}_\tau^{\mathrm{ex}})$ to $\mathcal{D}_{\mathrm{ex}}$
14:    **end if**
15: **end for**
16: Solve the IO training problem (3) with dataset $\mathcal{D}_{\mathrm{ex}}$ to obtain $\theta^*$
17: **Return:** policy parameters $\theta^*$

---

We then use Inverse Optimization to train a causal policy $\pi(s_t)$ that imitates the non-causal MPC expert policy (7), implicitly learning both the predictive relationship between past and future disturbances and the corresponding optimal response. The procedure used to approximate the non-causal MPC expert with IO is outlined in Algorithm (1).

**Remark 2.2** (Validity of in-hindsight trajectories)**.** The validity of this construction depends on the source of the mismatch. If disturbances are exogenous, i.e., generated by an external process independent of the state–action trajectory, then the non-causal MPC problem with in-hindsight disturbances (6) is equivalent to optimizing directly on the true system $f$, and the expert corresponds to the true optimizer. If disturbances depend on the state–action path, the disturbance sequence is path-dependent and cannot be reused counterfactually; in this case the in-hindsight expert remains a useful surrogate teacher, but not the true optimizer.

**Remark 2.3** (Literature on disturbance feedback and non-causal control)**.** The idea of "disturbance feedback control" has also been explored in recent works related to online control for adversarial disturbances (Hazan et al., 2020; Agarwal et al., 2019; Foster & Simchowitz, 2020). Additionally, a similar problem is also considered (Goel & Hassibi, 2021) where a non-causal controller is approximated by a causal one in an offline setting. Contrary to these works, which consider a linear policy class with no constraints on state or input, our proposed policy is nonlinear in nature and can handle constraints.

**Remark 2.4** (Feature engineering)**.** While the proposed quadratic hypothesis class (1) is affine in the feature space, it does not limit the policy to linear functions of the raw state. By selecting appropriate feature maps $\phi$ (e.g., polynomial expansions, sines/cosines of the state), one can model highly nonlinear control laws, as can be seen in the MuJoCo experiments Section 4.2. Additionally, we also empirically show that even simple features, such as disturbance histories, can be sufficient for even nonlinear control tasks (Section 4.1, Appendix B.3). Furthermore, for tasks requiring universal approximation capabilities, the proposed framework can be extended to use kernel methods (e.g., Gaussian kernels (Long et al., 2024) or Neural Tangent Kernels (Jacot et al., 2018)). This allows the algorithm to operate in high-dimensional implicit feature spaces without manual feature engineering, all while preserving the convexity of the training objective. In fact, in Section 4.2, we successfully employ the use of Gaussian kernels for certain experiments.

## 3 Robust Disturbance-Aware MPC

### 3.1 Robustification around disturbance trajectory

The non-causal MPC expert (7) optimizes directly against the noisy disturbance trajectory. However, due to stochasticity and/or potential distribution shifts in the data, performance might be degraded, and we may even observe instabilities. Therefore, we opt for a policy that is robust to such issues. To this end, let us introduce the robust counterpart to the non-causal MPC (7) described as

$$V_N^{\text{nc-rmpc}}(x, \mathbf{w}) := \min_{\mathbf{u}} \quad \max_{\bar{\mathbf{w}} \in \mathcal{W}(\mathbf{w})} \sum_{k=0}^{N-1} c(x_k, u_k) + c_f(x_N) \tag{8}$$
$$\text{s.t.} \quad \mathbf{u} \in \mathcal{U}_N^{\text{nc-rmpc}}(x, \mathbf{w}).$$

where $\mathcal{W}(\mathbf{w}) \subseteq \mathbb{R}^{Nn_w}$ is the disturbance uncertainty set centered around the trajectory $\mathbf{w}$, and

$$\mathcal{U}_N^{\text{nc-rmpc}}(x, \mathbf{w}) = \left\{ \mathbf{u} \in \mathbb{R}^{Nn_u} : \ \mathbf{u} \in \mathcal{U}_N^{\text{nc-mpc}}(x, \bar{\mathbf{w}}), \forall \bar{\mathbf{w}} \in \mathcal{W}(\mathbf{w}) \right\}. \tag{9}$$

A problem like (8) can easily be computationally intractable, even if its non-robust version (6) is not. When dealing with such problems, it is therefore common for conservative approximations to be used even when the nominal model $f_0$ is linear. Here, we propose an uncertainty set $\mathcal{W}$ for which (8) is tractable under linear dynamics and constraints and quadratic costs. Before we proceed, let us introduce a useful preparatory Lemma.

**Lemma 3.1** (Vectorized MPC formulation for linear dynamics)**.** *Under linear nominal dynamics $f_0(x, u) = Ax + Bu$, and quadratic costs $c(x, u) = \|x\|_{Q_x}^2 + \|u\|_{Q_u}^2$ and $c_f(x, u) = \|x\|_{Q_f}^2$, where $Q_x, Q_f \succcurlyeq 0$ and $Q_u \succ 0$, the objective of (6) can be equivalently expressed by*

$$\|\mathbf{A}x + \mathbf{B}\mathbf{u} + \mathbf{E}\mathbf{w}\|_{\mathbf{Q_x}}^2 + \|\mathbf{u}\|_{\mathbf{Q_u}}^2,$$

*with $\mathbf{Q_x} = \text{blkdiag}(I_{N-1} \otimes Q_x, Q_f)$, $\mathbf{Q_u} = I_N \otimes Q_u$, $\mathbf{A} = \text{blkcol}(A, \dots, A^N)$, $\mathbf{B} = \mathcal{T}_N(A, B)$, $\mathbf{E} = \mathcal{T}_N(A, E)$[2]. Moreover, when the stage constraints are polytopic $U = \{u \in \mathbb{R}^{n_u} : G_u u \le h_u\}$ and $X = \{x \in \mathbb{R}^{n_x} : G_x x \le h_x\}$, the constraint set of (6) is also polytopic in the form of*

$$\mathcal{U}_N^{\text{nc-mpc}}(x, \mathbf{w}) = \left\{ \mathbf{u} \in \mathbb{R}^{Nn_u} : \ \mathbf{F}x + \mathbf{G}\mathbf{u} \le \mathbf{h}(\mathbf{w}) \right\},$$

*with $\mathbf{F}^{\intercal} = \begin{bmatrix} (\mathbf{G_x}\mathbf{A})^{\intercal} & \mathbf{0} \end{bmatrix}$, $\mathbf{G}^{\intercal} = \begin{bmatrix} (\mathbf{G_x}\mathbf{B})^{\intercal} & \mathbf{G_u}^{\intercal} \end{bmatrix}$, $\mathbf{h}(\mathbf{w}) = \begin{bmatrix} (\mathbf{h_x} - \mathbf{G_x}\mathbf{E}\mathbf{w})^{\intercal} & \mathbf{h_u}^{\intercal} \end{bmatrix}$, $\mathbf{G_x} = I_N \otimes G_x$, $\mathbf{G_u} = I_N \otimes G_u$, $\mathbf{h_x} = \mathbf{1}_N \otimes h_x$, $\mathbf{h_u} = \mathbf{1}_N \otimes h_u$.*

Thanks to Lemma 3.1, the MPC problem (6) can be simplified to the convex quadratic program

$$\min_{\mathbf{u}} \quad \|\mathbf{A}x + \mathbf{B}\mathbf{u} + \mathbf{E}\mathbf{w}\|_{\mathbf{Q_x}}^2 + \|\mathbf{u}\|_{\mathbf{Q_u}}^2 \tag{10}$$
$$\text{s.t.} \quad \mathbf{F}x + \mathbf{G}\mathbf{u} \le \mathbf{h}(\mathbf{w})$$

The uncertainty set $\mathcal{W}$ we consider here is a ball centered on the $N$-length disturbance trajectory $\mathbf{w}$

$$\mathcal{W}(\mathbf{w}) := \left\{ \bar{\mathbf{w}} \in \mathbb{R}^{Nn_w} : \ \|\bar{\mathbf{w}} - \mathbf{w}\|_P^2 \le \varrho^2 \right\}, \tag{11}$$

where $P \succ 0$ is a desired geometry on the uncertainty trajectories. With this choice of uncertainty set, the robust constraints $\mathcal{U}_N^{\text{nc-rmpc}}(x, \mathbf{w})$, as defined in (9), enjoy an exact polytopic representation.

---

[2]Denotes a matrix $\mathcal{T}_N(A, B) = \begin{bmatrix} B & 0 & \cdots & 0 \\ AB & B & \cdots & 0 \\ \vdots & \vdots & \ddots & \vdots \\ A^{N-1}B & A^{N-2}B & \cdots & B \end{bmatrix}$.

**Lemma 3.2** (Exact polytopic representation of robust constraint set). *Under the hypotheses of Lemma 3.1 with uncertainty set (11) and $P \succ 0$, the constraints (9) have the following polytopic representation*

$$\mathbf{F}x + \mathbf{Gu} \leq \underline{\mathbf{h}}(\mathbf{w})$$

*where $\underline{\mathbf{h}}(\mathbf{w})^{\mathsf{T}} = \left[ (\mathbf{h_x} - \overline{g}(\mathbf{w}))^{\mathsf{T}} \quad \mathbf{h_u}^{\mathsf{T}} \right]$, $\overline{g}(\mathbf{w})^{\mathsf{T}} = \left[ \overline{g}_1(\mathbf{w}) \quad \overline{g}_2(\mathbf{w}) \quad \dots \right]$, and $\overline{g}_i(\mathbf{w}) = \varrho \left\| P^{-1/2} g_i \right\| + g_i^{\mathsf{T}} \mathbf{w}$, $\forall i$. The vectors $g_i$ are such that $[\mathbf{G_x E \bar{w}}]_i = g_i^{\mathsf{T}} \bar{\mathbf{w}}$.*

The proof is provided in Appendix A. We are now in a position to state our main result.

**Theorem 3.3** (Exact SDP reformulation). *Under the hypotheses of Lemmas 3.1 and 3.2, the robust non-causal MPC problem (8) is expressed as the min-max problem*

$$V_N^{\text{nc-rmpc}}(x, \mathbf{w}) = \min_{\mathbf{u}} \quad \max_{\bar{\mathbf{w}} \in \mathcal{W}(\mathbf{w})} \| \mathbf{A}x + \mathbf{Bu} + \mathbf{E\bar{w}} \|_{\mathbf{Q_x}}^2 + \| \mathbf{u} \|_{\mathbf{Q_u}}^2 \tag{12}$$
$$\text{s.t.} \quad \mathbf{F}x + \mathbf{Gu} \leq \underline{\mathbf{h}}(\mathbf{w})$$

*Furthermore, let us denote $\mathbf{X}(x, \mathbf{u}) = \mathbf{A}x + \mathbf{Bu}$. Then, the optimization problem (12) admits the convex reformulation*

$$\min_{\mathbf{u}, \lambda, \gamma_1, \gamma_2} \quad \gamma_1 + \gamma_2$$

$$\text{s.t.} \quad \lambda \geq 0, \quad \mathbf{F}x + \mathbf{Gu} \leq \underline{\mathbf{h}}(\mathbf{w}),$$

$$\begin{bmatrix} \mathbf{E}^{\mathsf{T}} \mathbf{Q_x E} - \lambda P & \mathbf{E}^{\mathsf{T}} \mathbf{Q_x X}(x, \mathbf{u}) + \lambda P \mathbf{w} \\ \star & -\gamma_1 - \lambda \left( \| \mathbf{w} \|_P^2 - \varrho^2 \right) \end{bmatrix} \preccurlyeq 0,$$

$$\begin{bmatrix} -I_N & \left( \mathbf{B}^{\mathsf{T}} \mathbf{Q_x B} + \mathbf{Q_u} \right)^{1/2} \mathbf{u} \\ \star & 2 \langle \mathbf{B}^{\mathsf{T}} \mathbf{Q_x A}x, \mathbf{u} \rangle + \| \mathbf{A}x \|_{\mathbf{Q_x}}^2 - \gamma_2 \end{bmatrix} \preccurlyeq 0.$$

The proof is relegated to Appendix A.

**Remark 3.4** (Uncertainty set). The uncertainty set (11) is not necessarily uniform in time as it is a ball on $Nn_w$-dimensional space, i.e., not all $\bar{w}_k$ components of $\bar{\mathbf{w}}$ need to be distanced equally from $w_k$. For instance, considering the case when $P = I_{Nn_w}$, we then have

$$\left\{ \bar{\mathbf{w}} \in \mathbb{R}^{Nn_w} : \textstyle\sum_{k=1}^N \| w_k - \bar{w}_k \|^2 \leq \varrho^2 \right\}.$$

The above uncertainty set includes disturbances with similar measures of energy to $\mathbf{w}$. Other similar approaches (Löfberg, 2003) aim to mitigate this by considering uncertainty sets such as $\{ \max_k \| w_k - \bar{w}_k \|^2 \leq \varrho^2 \}$, where each realization is bounded uniformly in time. However, since multiple quadratic inequalities are introduced as constraints, this necessitates the use of the inexact S-Lemma (Boyd et al., 1994), which inserts conservativeness. We use its exact version since only one quadratic inequality is involved in the constraints, thus allowing for an exact reformulation.

## 3.2 Approximating with Inverse Optimization

The non-causal policy (12) can be expressed in the form

$$\pi^{\text{nc-rmpc}}(x, \mathbf{w}) = \underset{u \in \mathcal{U}_1^{\text{nc-rmpc}}(x, \mathbf{w})}{\arg\min} \mathbf{Q}^{\text{nc-rmpc}}(x, u, \mathbf{w}) \tag{13}$$

with $\mathcal{U}_1^{\text{nc-rmpc}}(x, \mathbf{w})$ and $\mathbf{Q}^{\text{nc-rmpc}}(x, u, \mathbf{w})$ are defined accordingly, as in the previous sections. The procedure to approximate (13) with Inverse Optimization is identical to that used for the non-robust disturbance-aware MPC of Section 2.3 and is outlined by Algorithm 1. The only difference lies in the expert policy used; in this context, policy (13) is used instead of (7). One key difference with (7) is that (13) requires solving a semidefinite program –instead of a quadratic one– so we can expect greater computational improvement, albeit potentially at the expense of reducing the quality of the approximation.

By combining (3), and (7), we arrive at the convex optimization program whose solution is the fitted Q-function

$$\begin{cases} \min\limits_{\theta} & \sum\limits_{t=1}^{T} Q_\theta(\hat{s}_t, \hat{u}_t^{\mathrm{ex}}) - \min\limits_{u \in \mathcal{U}(s_t)} Q_\theta(\hat{s}_t, u) \\ \text{s.t.} & \hat{u}_t^{\mathrm{ex}} = \pi^{\mathrm{nc\text{-}rmpc}}(\hat{x}_t, \hat{\mathbf{w}}_{t+1}), \end{cases} \tag{14}$$

where the labels $\hat{u}_t^{\mathrm{ex}}$ are the in-hindsight optimal inputs computed by the min-max problem (8). An interesting parallel can be drawn between the exploration-exploitation dilemma and the robustification when computing the labels $\hat{u}_t^{\mathrm{ex}}$.

**Remark 3.5** (Exploration vs exploitation)**.** When looking at the exploration/exploitation dilemma as a competitive game between two conflicting objectives, we note that a similar trade-off exists in the min-max MPC (8) that is controlled by the uncertainty radius $\varrho$. This trade-off allows us to take into account disturbance trajectories different than the ones observed, a key feature that is addressed by exploration in RL and hence helps with generalization. This hypothesis is also confirmed by our numerical results in Section 4 and with additional experiments in the Appendix.

**Remark 3.6** (Computational considerations)**.** While the proposed Robust MPC is a semidefinite program (SDP), its complexity is only dependent on the planning horizon $N$ and the nominal model dimensions $(n_x, n_u)$, and does *not* scale with the dataset size $T$. Furthermore, the action improvement step can be computed independently for each data point, and is therefore trivially parallelizable. However, the IO distillation problem is also an SDP, with an LMI constraint for each sample; thus, the problem IO problem complexity does scale with the dataset size $T$. While modern solvers can handle large SDPs quite efficiently, we recognize that after a certain size, the IO problem might become intractable; in this case, iterative optimization methods may be employed (e.g., stochastic gradients (Zattoni Scroccaro et al., 2025), or block coordinate descent (Long et al., 2024)), which scale gracefully while retaining the theoretical guarantees of the convex landscape.

## 4 Numerical Experiments

In our numerical analysis, we focus on two domains, the quadrotor environment from safe-control-gym (Brunke et al., 2021) and the MuJoCo control benchmark (Todorov et al., 2012). Additionally, we include detailed ablation studies for two more experiments found in Appendix Section B: the control of the linearized dynamics of a fighter jet (Safonov et al., 1981) and a nonlinear temperature control problem.

### 4.1 Quadrotor environment

We conduct experiments in a nonlinear quadrotor environment from safe-control-gym (Brunke et al., 2021) and evaluate our approach in comparison with two RL algorithms: Proximal Policy Optimization (PPO) (Schulman et al., 2017) and Conservative Q-Learning (CQL) (Kumar et al., 2020). Both are model-free, with CQL falling under the offline RL paradigm and PPO being an on-policy algorithm.

**Environment specifications:** The quadrotor environment consists of a 6-dimensional state space and two control inputs. The objective is to reach a fixed goal state starting from a randomly sampled starting position while keeping the quadrotor stable under an unknown external force that acts as a disturbance. This force consists of a sinusoidal signal of a random phase with additive Gaussian noise applied to the body of the quadrotor. The environment has a nonlinear dynamical system, assumed to be known, with the minimum and maximum episodic cost being 0 and 300, respectively.

**Experimental Setup:** We linearize the dynamics around an equilibrium point to form a nominal model before giving it to the MPC policies. In the following experiments, we denote an MPC policy that is oblivious to the external sinusoidal disturbance by MPC (obl), and with MPC (f-dst), we refer to an MPC that has the full information of the future disturbance trajectory. In our evaluations, we trained the PPO agent with 3M environment steps, which we refer to as PPO-3M, and the CQL agent for 50k iterations, where we observed convergence in performance. Both IO and CQL agents are trained with the same dataset generated by an MPC (obl) policy with a 25-step horizon.

Table 1: Distribution (mean ± standard deviation over 4 seeds) of last-epoch scores of different baselines in the MuJoCo benchmark. All datasets are "medium". The "Dataset" column represents the performance of the trajectories used for training. The suffix in the environment name (e.g., 5K, 10K, 1M) denotes the amount of data used, obtained by taking the first $N$ chunks of the dataset.

| Environment | Dataset | IO | IQL | CQL | COMBO | MOPO | TD3BC |
|---|---|---|---|---|---|---|---|
| `hopper` 5K | 47.2±19.2 | 45.9±3.2 | 46.8±3.6 | 49.0±4.3 | 50.9±1.1 | 4.1±4.0 | 12.5±9.8 |
| `hopper` 1M | 44.3±11.6 | 51.7±0.9 | 65.7±8.1 | 59.1±4.1 | 84.7±9.3 | 62.8±38.1 | 60.8±3.4 |
| `walker2d` 10K | 65.9±17.9 | 42.2±1.7 | 54.4±7.0 | 51.0±7.0 | 40.7±25.7 | 5.0±9.1 | 1.1±1.0 |
| `walker2d` 1M | 62.1±23.9 | 71.8±1.9 | 81.1±2.6 | 83.6±0.5 | 83.9±1.9 | 85.4±2.9 | 84.4±2.1 |

Table 2: Distribution (mean ± standard deviation) of best-epoch scores of different baselines in the MuJoCo benchmark. All datasets are "medium". The "Dataset" column represents the performance of the trajectories used for training. The suffix in the environment name (e.g., 5K, 10K, 1M) denotes the amount of data used, obtained by taking the first $N$ chunks of the dataset.

| Environment | Dataset | IO | IQL | CQL | COMBO | MOPO | TD3BC |
|---|---|---|---|---|---|---|---|
| `hopper` 5K | 47.2±19.2 | 83.8±9.7 | 90.5±7.1 | 72.3±1.5 | 68.4±7.1 | 26.2±8.4 | 32.6±0.9 |
| `hopper` 1M | 44.3±11.6 | 60.8±3.2 | 95.4±2.1 | 89.3±2.8 | 100.2±0.4 | 93.1±27.9 | 77.0±1.7 |
| `walker2d` 10K | 65.9±17.9 | 72.1±1.4 | 73.9±1.6 | 67.8±2.2 | 76.3±3.0 | 16.9±4.6 | 5.4±0.9 |
| `walker2d` 1M | 62.1±23.9 | 77.5±0.8 | 87.4±0.5 | 87.7±0.5 | 87.4±0.6 | 92.5±0.8 | 87.4±0.7 |

Figures 1 and 2 show our comparisons and ablation studies in the quadrotor environment. In all six figures, $T$ denotes the dataset length, $N$ is the MPC horizon, and $H$ is the lookback horizon. IO-RMPC* is the $\rho$-tuned policy. Unless stated otherwise, the default values of $N$ and $H$ are set to 25 and 2, respectively, and each evaluation of an agent is performed with 20 different starting points. To normalize the effect of the randomized initial starting points, in both figures, we only report the steady-state[3] costs. The dashed lines indicate the median values, and the tubes contain the range between the 20th to 80th percentiles of the costs, if not stated otherwise.

**Comparisons:** In the left plot of Fig. 1, we compare the episodic cost histograms of four agents evaluated with 20 different initial conditions. Our evaluations show that IO-RMPC yields significantly lower costs even with a limited dataset of $T = 3,000$ samples. The center plot of Fig. 1 shows a comparison of the IO-RMPC policy against various CQL agents. Although CQL converges to IO-RMPC performance, it requires an order of magnitude more samples. Finally, we compare the MPC performances with the PPO agent. Although MPC policies are only given a linear nominal model of the environment, starting with the 15-step horizon, they surpass the PPO performance.

**Ablation studies:** Additionally, we analyze the effect of the uncertainty radius $\rho$ and lookback horizon $H$. The center plot of Fig. 2 indicates that robustification of the IO-MPC policy, up to some value of $\rho$, improves performance even in the absence of a disturbance bias. This behavior is also present in the left plot of Fig. 2, where the IO-RMPC policy even surpasses the MPC (f-dst) policy. We posit that this is due to the fact that robustifying also helps with model mismatch between the actual dynamics and the nominal one. Finally, we perform an ablation on the lookback horizon of the IO-MPC policy shown in the right plot of Fig. 2. We observe that when the parameter $H$ is set to 2, IO-MPC almost recovers the performance of the full-information MPC (f-dst) policy, whereas a further increase in $H$ degrades performance.

## 4.2 MuJoCo benchmark

Next, we compare IO agents with widely used model-based and model-free offline RL algorithms within the MuJoCo control benchmark (Todorov et al., 2012). In these experiments, we employ a model-free version of the IO agent, where the actions $\hat{u}_t^{\mathrm{ex}}$ in Algorithm (1) are directly taken from the dataset. The augmented

---

[3]Defined as the last 40% of data points of a trajectory.

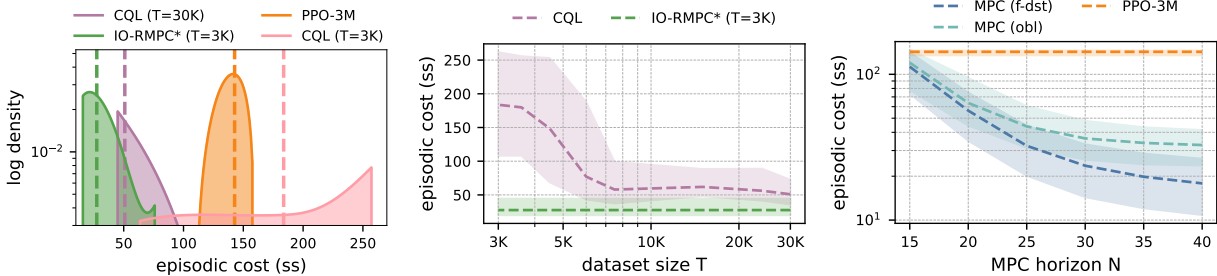

Figure 1: Comparisons of several agents in the quadrotor environment. **Left:** The cost histogram of the offline IO and CQL agents and online model-based MPC and model-free PPO-3M (trained with 3M environment steps) agents. **Center:** The cost distributions of CQL agents trained with 4 seeds on various dataset lengths compared to a single IO-RMPC policy trained with 3000 samples. **Right:** Comparison of the cost distributions between oblivious and full disturbance MPC policies against the model-free PPO agent.

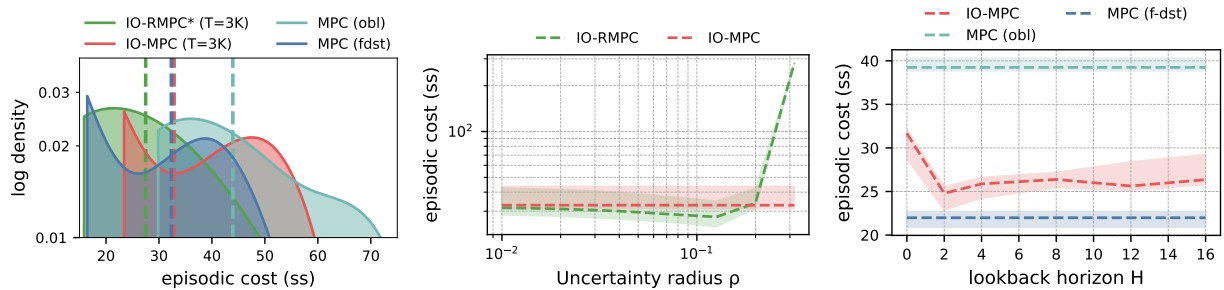

Figure 2: Ablation studies of MPC and IO policies in the quadrotor environment. **Left:** The cost histogram of IO and MPC agents with a 25-step horizon. **Center:** The cost distribution of IO-RMPC policy with different $\rho$ values and IO-MPC policy with the same horizon $N$. The tube contains the range from the 40th to the 60th percentiles of the costs. **Right:** The steady-state cost distributions of the IO-MPC policy with various look-back horizons ($H$) against MPC policies. The tube contains a narrower range from the 45th to the 55th percentiles of the costs.

Table 3: Comparison of parameter counts and per-epoch wall-clock time (elapsed time per training epoch) in `walker2d` across algorithms. Wall-clock time is measured on an NVIDIA GeForce RTX 3090 GPU.

|                     | IO    | IQL     | CQL     | COMBO     | MOPO      | TD3BC   |
|---------------------|-------|---------|---------|-----------|-----------|---------|
| Parameters          | 9,390 | 286,214 | 414,732 | 1,320,672 | 1,123,296 | 215,814 |
| Train time (1 epoch)| 1.24s | 160.97s | 418.11s | 479.86s   | 191.55s   | 96.07s  |

state $\phi(\hat{\mathbf{x}}_{t-4:t}, \hat{\mathbf{u}}_{t-4:t})$ includes the last four state-action pairs, the cross-products of state features, a constant bias term, Radial Basis Function (RBF) features over the state, and the state sinusoidal terms. The latter augmentation is motivated by the periodic nature of the targeted tasks in robotics.

**Experiment setup:** We use the dataset from the D4RL repository (Fu et al., 2020) to train IO agents and offline RL algorithms. We employ an iterative version of the IO algorithm, using gradient-based optimization to minimize the objective function in Equation (2). We trained each algorithm with four different seeds and evaluated the agents after each epoch using 40 different seeds throughout the training process. In addition to experiments with the full dataset (1M samples), we also evaluate low-data regimes by restricting the training set to the first 10K samples for `walker2d` and the first 5K samples for `hopper`. We report the evaluation scores at last-epoch of the training in Table 1, and best-epoch evaluation scores[4] across training in Table 2.

---

[4]We excluded the RBF features in the low-data regime runs of the IO agent. See Table 4 in Appendix for details.

We note that selecting the first 5K or 10K samples is arbitrary, and that the average dataset score is largely unchanged relative to the full dataset. We obtained the scores for the offline RL algorithms by running the implementations provided in the OfflineRL-Kit repository (Sun, 2023), which match the originally reported scores when the algorithms are executed on the full dataset. See Appendix B.1 for a detailed study with the IO agent across varying sizes of uniformly sampled training sets.

The best-epoch scores provided in Table 2 demonstrate the expressiveness of the IO hypothesis class in the low-data regime of both the `hopper` and `walker2d` environments. In that regime, the IO agent achieves competitive scores compared to widely used model-based and model-free baselines. Using the last-epoch scores in Table 1, we show that the IO agent attains dataset-level performance when trained on the full dataset while using an order of magnitude fewer parameters; Table 3 reports parameter counts and per-epoch wall-clock time (measured in `walker2d`) across algorithms. This outcome is expected, since the IO agent does not employ the action improvement step proposed in Section 3 in the D4RL experiments.

We argue that the successful performance of the IO algorithm with such a low number of parameters is due to the inherent richness of the IO hypothesis class, combined with a convex optimization loss function that allows us to provably reach the (in-sample) global optimizer during the training phase. Furthermore, due to the inherent simplicity of the proposed policy class, the IO algorithm is able to generalize with significantly fewer samples.

In these experiments, we refrain from running our proposed IO-RMPC agent that employs the action improvement step, since constructing a nominal model for MuJoCo tasks, required for the MPC experts, is a task that is inherently difficult and beyond the scope of this work. Nevertheless, our experiments with the plain IO agent reveal promising and competitive results in MuJoCo control tasks. These results underscore the substantial potential of IO-based algorithms within the RL or IL contexts, especially in scenarios with limited data. Extending the RMPC-based action improvement step to deal with more complicated dynamics remains an avenue for future research.

## 5 Concluding Remarks, Limitation, and Future Directions

In this work, we presented a convex and robust offline RL framework that utilizes a nominal model and in-hindsight information to learn an optimal policy. Through empirical evaluations, we showcased that our proposed algorithm can recover the performance of non-causal agents with complete environmental knowledge, while at the same time significantly outperforming RL algorithms in the low-sample data regimes (both online and offline). We further demonstrated that the IO framework, leveraging its expressivity and convexity properties, effectively recovers teacher-level performance in challenging MuJoCo offline control tasks. Our results show that IO yields performance comparable to established reward-driven baselines, particularly in low-data regimes, while employing orders of magnitude fewer parameters than its competitors.

We also find it essential to mention some of the inherent limitations of our approach. While the proposed quadratic hypothesis class, when paired with appropriate features, has demonstrated sufficient expressiveness in the control environments examined within our numerical studies, for more sophisticated tasks, additional steps can be required, such as applying kernel tricks, as was done for some of the MuJoCo experiments, or employing a nonlinear state embedding. Another drawback of our approach is the reliance of our robust MPC formulation on a nominal model. This requirement can become impractical for complex environments where approximating a nominal model is challenging. However, these limitations are not inherent and can be potential avenues for future research, including topics such as:

(i) approximating non-causal policies by utilizing in-hindsight information in real-time, using tools from Online Convex Optimization; and

(ii) extending the robust min-max optimization (RMPC) framework to off-policy and offline RL settings.

As we conclude, we position our approach as a step towards bridging the gap between robust control and offline RL, offering a particular applicability in continuous control tasks with substantial distribution shifts from training to test and also in environments where the availability of training data is limited.

## Acknowledgements

The authors would like to thank the reviewers for their useful comments. This work was partially supported by the Horizon Europe Pathfinder Open project RELIEVE-101099481 and by the European Research Council (ERC) project TRUST-949796.

## A Technical Proofs

### A.1 Proof of Lemma 3.2

The original constraint expresses a row-wise inequality. With the parameterization $[\mathbf{G_x E \bar{w}}]_i = g_i^\intercal \bar{\mathbf{w}}$, the inequality $\mathbf{F}x + \mathbf{Gu} \leq \mathbf{h}(\bar{\mathbf{w}})$, $\forall \bar{\mathbf{w}} \in \mathcal{W}(\mathbf{w})$ is equivalent to solving the following optimization program for every $i$:

$$\overline{g}_i(\mathbf{w}) = \max_{\bar{\mathbf{w}}} \left\{ g_i^\intercal \bar{\mathbf{w}} : \ \|\bar{\mathbf{w}} - \mathbf{w}\|_P^2 \leq \varrho^2 \right\}$$

To that end, let $\tilde{\mathbf{w}} = \varrho^{-1} P^{1/2}(\bar{\mathbf{w}} - \mathbf{w})$. Then the above becomes

$$\overline{g}_i(\mathbf{w}) = \max_{\tilde{\mathbf{w}}} \left\{ g_i^\intercal (\varrho P^{-1/2} \tilde{\mathbf{w}} + \mathbf{w}) : \ \|\tilde{\mathbf{w}}\| \leq 1 \right\}$$

The maximization of a linear function on the unit disk has an analytical solution and that is

$$\overline{g}_i(\mathbf{w}) = \varrho \left\| P^{-1/2} g_i \right\| + g_i^\intercal \mathbf{w}$$

By putting everything together we conclude the proof.

### A.2 Proof of Theorem 3.3

The program (12) follows directly by combining the results of Lemmas 3.1 and 3.2. Let us denote the inner maximization as

$$J(\mathbf{u}) := \max_{\bar{\mathbf{w}} \in \mathcal{W}(\mathbf{w})} \ \|\mathbf{A}x + \mathbf{Bu} + \mathbf{E}\bar{\mathbf{w}}\|_{\mathbf{Q_x}}^2 + \|\mathbf{u}\|_{\mathbf{Q_u}}^2$$

and its corresponding Lagrangian as

$$\mathcal{L}_J(\lambda, \mathbf{u}, \bar{\mathbf{w}}) := \|\mathbf{A}x + \mathbf{Bu} + \mathbf{E}\bar{\mathbf{w}}\|_{\mathbf{Q_x}}^2 + \|\mathbf{u}\|_{\mathbf{Q_u}}^2 - \lambda \left( \|\bar{\mathbf{w}} - \mathbf{w}\|_P^2 - \varrho^2 \right)$$

After some manipulations and rearrangements, we have

$$\mathcal{L}_J(\lambda, \mathbf{u}, \bar{\mathbf{w}}) = \langle \bar{\mathbf{w}}, \left( \mathbf{E}^\intercal \mathbf{Q_x E} - \lambda P \right) \bar{\mathbf{w}} \rangle + 2 \langle \mathbf{E}^\intercal \mathbf{Q_x} \left( \mathbf{A}x + \mathbf{Bu} \right) + \lambda P \mathbf{w}, \bar{\mathbf{w}} \rangle$$
$$+ \|\mathbf{A}x + \mathbf{Bu}\|_{\mathbf{Q_x}}^2 + \|\mathbf{u}\|_{\mathbf{Q_u}}^2 - \lambda \left( \|\mathbf{w}\|_P^2 - \varrho^2 \right)$$

Let us introduce the following notation

$$\Lambda(\lambda) := \mathbf{E}^\intercal \mathbf{Q_x E} - \lambda P$$
$$M(\lambda, \mathbf{u}) := \mathbf{E}^\intercal \mathbf{Q_x} \left( \mathbf{A}x + \mathbf{Bu} \right) + \lambda P \mathbf{w}$$
$$\nu_1(\lambda) := -\lambda \left( \|\mathbf{w}\|_P^2 - \varrho^2 \right)$$
$$\nu_2(\mathbf{u}) := \|\mathbf{A}x + \mathbf{Bu}\|_{\mathbf{Q_x}}^2 + \|\mathbf{u}\|_{\mathbf{Q_u}}^2$$
$$\nu(\lambda, \mathbf{u}) := \nu_1(\lambda) + \nu_2(\mathbf{u})$$

The dual of this problem is then

$$d_J(\lambda, \mathbf{u}) := \max_{\bar{\mathbf{w}}} \mathcal{L}_J(\lambda, \mathbf{u}, \bar{\mathbf{w}}) = \begin{cases} -M(\lambda, \mathbf{u})^\intercal \Lambda(\lambda)^\dagger M(\lambda, \mathbf{u}) + \nu(\lambda, \mathbf{u}), \\ \quad \text{if } \Lambda(\lambda) \preccurlyeq 0 \text{ and } M(\lambda, \mathbf{u})^\intercal \left( I - \Lambda(\lambda)\Lambda(\lambda)^\dagger \right) = 0 \\ +\infty, \text{ otherwise} \end{cases}$$

Strong duality holds due to the S-Lemma Boyd & Vandenberghe (2004). Therefore, $J(\mathbf{u}) = \min_{\lambda \geq 0} d_J(\lambda, \mathbf{u})$. Now consider the following epigraph reformulation

$$
\begin{aligned}
J(\mathbf{u}) = \min_{\lambda, \gamma_1} \quad & \gamma_1 + \nu_2(\mathbf{u}) \\
\text{s.t.} \quad & \lambda \geq 0, \\
& \Lambda(\lambda) \preccurlyeq 0, \\
& M(\lambda, \mathbf{u})^{\mathsf{T}} \left( I - \Lambda(\lambda)\Lambda(\lambda)^{\dagger} \right) = 0, \\
& - M(\lambda, \mathbf{u})^{\mathsf{T}} \Lambda(\lambda)^{\dagger} M(\lambda, \mathbf{u}) + \nu_1(\lambda) \leq \gamma_1
\end{aligned}
$$

The last three constraints can be cast as an LMI using the non-strict Schur complement Boyd et al. (1994) and we have

$$
\begin{aligned}
J(\mathbf{u}) = \min_{\lambda, \gamma_1} \quad & \gamma_1 + \nu_2(\mathbf{u}) \\
\text{s.t.} \quad & \lambda \geq 0, \\
& \begin{bmatrix} \Lambda(\lambda) & M(\lambda, \mathbf{u}) \\ \star & \nu_1(\lambda) - \gamma_1 \end{bmatrix} \preccurlyeq 0
\end{aligned}
$$

Therefore the overall robust MPC problem can now be written as $\min_{\mathbf{u}} \{ J(\mathbf{u}) : \mathbf{F}x + \mathbf{G}\mathbf{u} \leq \underline{\mathbf{h}}(\mathbf{w}) \}$. In order to write this in the standard SDP form, we will have to use another epigraph reformulation, that of $\nu_2(\mathbf{u})$:

$$
\begin{aligned}
\min_{\lambda, \gamma_1, \gamma_2} \quad & \gamma_1 + \gamma_2 \\
\text{s.t.} \quad & \lambda \geq 0, \\
& \begin{bmatrix} \Lambda(\lambda) & M(\lambda, \mathbf{u}) \\ \star & \nu_1(\lambda) - \gamma_1 \end{bmatrix} \preccurlyeq 0, \\
& \mathbf{F}x + \mathbf{G}\mathbf{u} \leq \underline{\mathbf{h}}(\mathbf{w}), \\
& \|\mathbf{A}x + \mathbf{B}\mathbf{u}\|_{\mathbf{Q_x}}^2 + \|\mathbf{u}\|_{\mathbf{Q_u}}^2 \leq \gamma_2
\end{aligned}
$$

The last constraint can now be written as $\gamma_2 \geq \langle \mathbf{u}, (\mathbf{B}^{\mathsf{T}} \mathbf{Q_x} \mathbf{B} + \mathbf{Q_u}) \mathbf{u} \rangle + 2 \langle \mathbf{B}^{\mathsf{T}} \mathbf{Q_x} \mathbf{A}x, \mathbf{u} \rangle + \|\mathbf{A}x\|_{\mathbf{Q_x}}^2$, which can be expressed as the LMI:

$$
\begin{bmatrix} -I_N & (\mathbf{B}^{\mathsf{T}} \mathbf{Q_x} \mathbf{B} + \mathbf{Q_u})^{1/2} \mathbf{u} \\ \star & 2 \langle \mathbf{B}^{\mathsf{T}} \mathbf{Q_x} \mathbf{A}x, \mathbf{u} \rangle + \|\mathbf{A}x\|_{\mathbf{Q_x}}^2 - \gamma_2 \end{bmatrix} \preccurlyeq 0
$$

Hence, by putting everything together we arrive that the original problem (12) is equivalent to:

$$
\begin{aligned}
\min_{\mathbf{u}, \lambda, \gamma_1, \gamma_2} \quad & \gamma_1 + \gamma_2 \\
\text{s.t.} \quad & \lambda \geq 0, \\
& \mathbf{F}x + \mathbf{G}\mathbf{u} \leq \underline{\mathbf{h}}(\mathbf{w}), \\
& \begin{bmatrix} \mathbf{E}^{\mathsf{T}} \mathbf{Q_x} \mathbf{E} - \lambda P & \mathbf{E}^{\mathsf{T}} \mathbf{Q_x} (\mathbf{A}x + \mathbf{B}\mathbf{u}) + \lambda P\mathbf{w} \\ \star & -\gamma_1 - \lambda \left( \|\mathbf{w}\|_P^2 - \varrho^2 \right) \end{bmatrix} \preccurlyeq 0, \\
& \begin{bmatrix} -I_N & (\mathbf{B}^{\mathsf{T}} \mathbf{Q_x} \mathbf{B} + \mathbf{Q_u})^{1/2} \mathbf{u} \\ \star & 2 \langle \mathbf{B}^{\mathsf{T}} \mathbf{Q_x} \mathbf{A}x, \mathbf{u} \rangle + \|\mathbf{A}x\|_{\mathbf{Q_x}}^2 - \gamma_2 \end{bmatrix} \preccurlyeq 0.
\end{aligned}
$$

We have arrived at the formulation in the Theorem statement, and as such, we conclude the proof.

## B   Additional Numerical Experiments

Besides the numerical experiments in Section 4, here we include more details and numerical results for the MuJoCo experiments of Section 4.2, and we further include two more examples that enable us to study our approach in more detail.

Table 4: IO hyperparameters. The suffix in the environment name (e.g., last, best) denotes the runs to produce the respective scores.

| Environment | epoch | lr | lr decay | RBF |
|---|---|---|---|---|
| walker2d 10K (last) | 400 | 0.05 | 0.975 | Yes |
| walker2d 10K (best) | 100 | 0.05 | 0.985 | No |
| walker2d 1M | 400 | 0.05 | 0.975 | Yes |
| hopper 5K | 100 | 0.05 | 0.9625 | No |
| hopper 1M | 100 | 0.05 | 0.95 | Yes |

Table 5: The IO agent performance in `walker2d-medium` using a subset of uniformly sampled datasets.

| Datasize | Last | Last 5% | Best | Best 5% |
|---|---|---|---|---|
| 10K | 40.7±2.0 | 41.5±0.7 | 55.0±1.4 | 49.9±0.5 |
| 50K | 50.2±1.9 | 48.9±0.7 | 63.2±1.9 | 57.6±0.8 |
| 250K | 53.0±1.6 | 52.4±1.2 | 68.0±3.2 | 62.4±0.6 |
| 1M | 71.8±1.9 | 70.8±0.6 | 77.5±0.8 | 75.6±0.3 |

## B.1 MuJoCo − additional results

Here, we provide details of the runs reported in Section 4.2 and ablation studies on the D4RL benchmark.

We obtain the scores in Table 2 and Table 1 by running each algorithm for 100 epochs with 10,000 gradient update steps per epoch, except for MOPO (300 epochs) and the full-dataset (1M samples) IO experiments (400 epochs). Table 4 reports the number of epochs used for the IO experiments, along with other hyperparameters, across all dataset configurations.

In the experiments reported in Section 4.2, we use a fixed chunk from each dataset: the first 5K samples for `hopper` and the first 10K samples for `walker2d`. Additionally, Table 5 shows how IO performance scales with the size of a uniformly sampled training set in `walker2d`. We observe that both last-epoch and best-epoch performance improve as the dataset size increases. Moreover, the `walker2d` score obtained using the first 10K samples (Table 1) is close to the score obtained using a uniformly sampled 10K subset.

In addition to the ablation on dataset size, we compare the performance of the IO agent across two different quality levels of the `walker2d` dataset, namely "medium" and "expert". Table 6 reports the scores for both settings. Comparing the "expert" and "medium" results, we observe that the performance of the IO agent scales with the quality of the input data. The best-epoch scores show that the agent is capable of exceeding the teacher in both cases (110 for "expert", 77 for "medium"). However, the "medium" dataset shows significantly higher stability at the end of training (a smaller gap between best and last), suggesting that the "medium" distribution may be easier for the quadratic hypothesis class to represent robustly without overfitting.

In Figure 3, we show the evaluation scores at each epoch for all algorithms, along with the dataset average, for both environments (i.e., `hopper` and `walker2d`) in the low-data and full-data regimes. The figure corresponds to the runs reported in Table 1. For ease of visualization and comparison, although we run MOPO (3M steps) and IO (see Table 4) for more than 1M steps, we plot only the first 1M steps (100 epochs). Overall, the results suggest that, in the low-data regime, the IO agent performs competitively against the best offline RL baselines, while in the full-data regime it can exceed the teacher (dataset average) score.

## B.2 Linear fighter jet

We consider the regulation of the unstable dynamics of a six-dimensional fighter jet (Safonov et al., 1981) with additive unknown disturbances $w_{t+1} = f_w(t; w_0) + v_{t+1}$, where $f_w$ has a sinusoidal component with random phase $w_0 \sim \mathcal{U}[0, \pi/2]$ and a bias term, and $v_t \sim \mathcal{N}(0, \Sigma_v)$. As the dynamics are given and linear, the nominal model $\tilde{f}_0(x, u, w) = Ax + Bu + Ew$ coincides with the true dynamics $f$. Initial conditions are sampled

Table 6: The IO agent performance in `walker2d` with full dataset (1M samples) across "medium" and "exper" data qualities.

| Experiment | Last | Last 5% | Best | Best 5% |
|---|---|---|---|---|
| `walker2d-expert` | 77.0±8.2 | 77.5±3.4 | 109.7±0.2 | 108.5±0.2 |
| `walker2d-medium` | 71.8±1.9 | 70.8±0.6 | 77.5±0.8 | 75.6±0.3 |

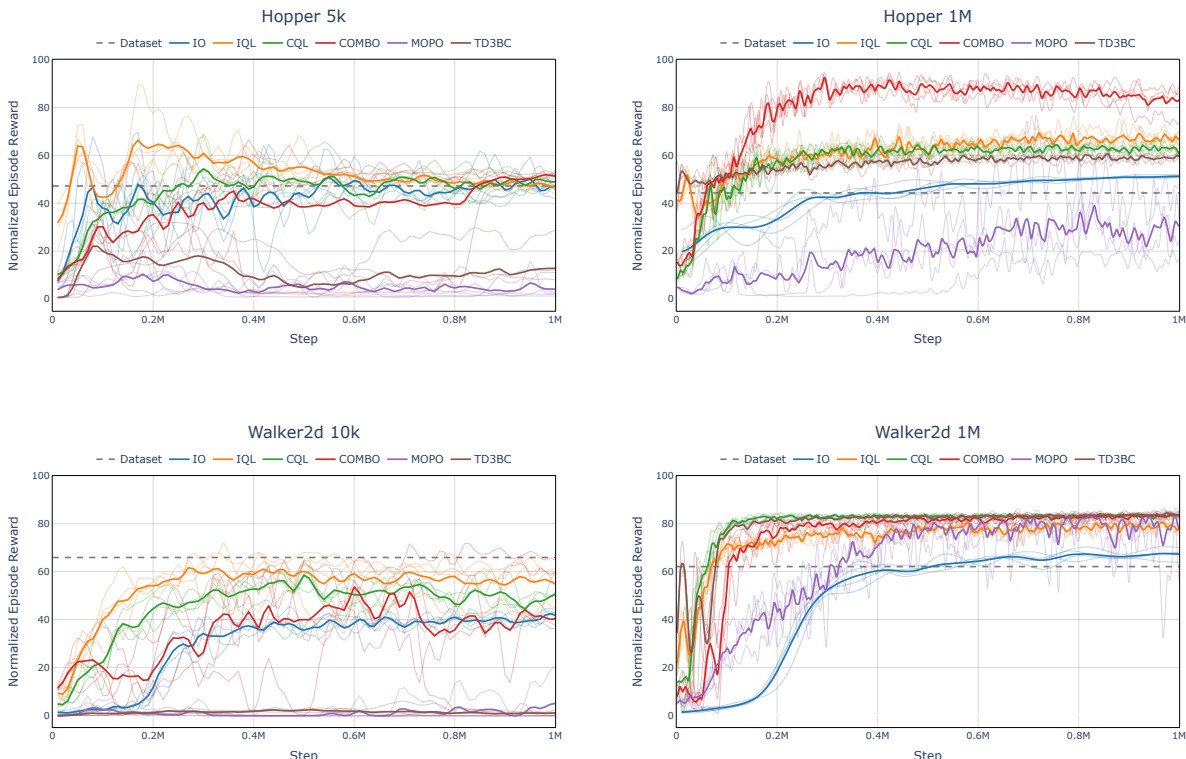

Figure 3: Normalized episodic reward of all agents, including the dataset average (gray dashed line), across 1M steps (100 epochs): `hopper` using (top left) the first 5K samples and (top right) the full dataset, and `walker2d` using (bottom left) the first 10K samples and (bottom right) the full dataset. Thick curves represent the mean reward, and the transparent curves show the reward of individual seeds, averaged over 40 evaluations at each epoch. The curves are smoothed for clarity.

randomly as $x_0 \sim \mathcal{N}(0, 0.1I_6)$. Further, we impose that the state be constrained in $\{x \in \mathbb{R}^6 : |x^1| \leq 1\}$ and the input in $\{u \in \mathbb{R}^2 : |u^1| \leq 2, |u^2| \leq 3\}$. We select the IO features as $\phi(\mathbf{x}_{1:t}, \mathbf{u}_{1:t}) = (x_t, 1, w_{t-1}, w_t)$.

The dynamics of the fighter jet Safonov et al. (1981) have been discretized with a sampling time of 0.035 s, resulting in the following discrete-time system matrices:

$$
A = \begin{bmatrix}
0.9991 & -1.3736 & -0.6730 & -1.1226 & 0.3420 & -0.2069 \\
0.0000 & 0.9422 & 0.0319 & -0.0000 & -0.0166 & 0.0091 \\
0.0004 & 0.3795 & 0.9184 & -0.0002 & -0.6518 & 0.4612 \\
0.0000 & 0.0068 & 0.0335 & 1.0000 & -0.0136 & 0.0096 \\
0 & 0 & 0 & 0 & 0.3499 & 0 \\
0 & 0 & 0 & 0 & 0 & 0.3499
\end{bmatrix}, B = \begin{bmatrix}
0.1457 & -0.0819 \\
-0.0072 & 0.0035 \\
-0.4085 & 0.2893 \\
-0.0052 & 0.0037 \\
0.6501 & 0 \\
0 & 0.6501
\end{bmatrix}, E = \begin{bmatrix}
0 & 0 \\
0 & 0 \\
1 & 0 \\
0 & 1 \\
0 & 0 \\
0 & 0
\end{bmatrix}.
$$

As mentioned in the main body, the disturbances are $w_{t+1} = f_w(t; w_0) + v_{t+1}$, where $v_t \sim (0, \Sigma v)$, $w_0 \sim \mathcal{U}[0, \pi/2]$, with

$$f_w(t; w_0) = \begin{bmatrix} 0.5\sin(4.488t + w_0) \\ 0.01 \end{bmatrix} \text{ and } \Sigma_v = \begin{bmatrix} 0.01 & 0 \\ 0 & 0.001 \end{bmatrix}.$$

The cost parameters are selected as $Q_f = Q_x = \mathrm{diag}(1, 10^3, 10^2, 10^3, 1, 1)$ and $Q_u = I_2$, and the MPC horizon is $N = 20$.

**Approximating NC-MPC with IO:** First, we want to validate that hindsight can be used to mitigate unknown disturbances. As such, we will compare the following policies: **MPC (obl)**, an MPC that can measure only $x_t$ at time $t$ and does not know $f_w$, as described in (4); **MPC (dst)**, an MPC that can measure both $x_t$ and $w_{t+1}$ at time $t$, and also knows $f_w$; and **IO-MPC**, the policy resulting from applying Algorithm 1 to a dataset of trajectories obtained from MPC (obl). All IO-derived policies described in this paragraph and the next are trained with a dataset containing 10 trajectories induced by MPC (obl) of length 51 each. In the left plot of Figure 4 we have the cost histogram of $c(x, u)$ for each tested policy during steady state[5]. We can see that in both plots the IO-MPC policy recovers a significant part of the performance of MPC (dst), both in terms of median and of variance.

**Approximating NC-RMPC with IO:** Using the same setup and data, we impose a distribution shift in the disturbances during evaluation by adding a constant bias to $w_t$; specifically, we apply $\tilde{w}_t$ instead of $w_t$, where $\tilde{w}_t^\intercal = w_t^\intercal + \begin{bmatrix} 0.1 & 0.05 \end{bmatrix}^\intercal$. We therefore compare the following: **MPC (obl)**, as before; **MPC (p-dst)**, as MPC (dst) of the previous section – only measures $w_{t+1}$; **MPC (f-dst)**, similar to MPC (p-dst), except that it has access to $\tilde{w}_{t+1}$ instead of $w_{t+1}$; **IO-MPC**, as before; **IO-RMPC**, a robust MPC of the form (12), trained with the same data as IO-MPC and equipped with $P = I_{Nn_w}$ and $\varrho = 10^{-2}$. It is immediately obvious from the middle and rightmost cost distributions of Figure 4 that imitating the robust expert yields performance benefits when faced with distribution shift, as the median performance of IO-RMPC is better than that of IO-MPC. Not only that, but IO-RMPC manages to recover the median performance of MPC (f-dst), albeit with a larger variance.

**Effect of uncertainty radius:** We further explore the impact of the robustness parameter (uncertainty radius $\varrho$) on the steady-state cost distribution across different training datasets. In the left plot of Fig. 5, we observe that increasing $\varrho$ until $\varrho^*$ yields a consistent reduction in the time-averaged steady-state cost across different training sets. What is surprisingly interesting is that there are some datasets which, when trained with properly tuned $\varrho$, can match the performance of the full-information agent MPC (f-dst). We also looked into the performance of such controllers on the entire distribution of the steady-state cost in the middle plot of Fig. 5: $\varrho$ has a positive impact on the entire steady-state cost distribution (and not only the median or average). We also note that the non-robust controller IO-MPC coincides with the robust one (IO-RMPC) for sufficiently small $\varrho$. In the right plot of Fig. 5, we freeze $\varrho = \varrho^*$ and look at the entire steady-state cost distributions of the three policies involved in the middle plot. We observe that even though the median performance of IO-RMPC surpasses that of MPC (f-dst), its variance across the test set is much more spread, making it more high-risk than MPC (f-dst). However, as the variance of the non-robust IO policy is similarly wide, the takeaway message here is that robustification combats distribution shift during policy evaluation.

## B.3 Nonlinear temperature control

Here, we consider a nonlinear 4-th order dynamical system that describes the heat transfer equations of two coupled heating elements (inputs) and two temperature sensors (outputs), akin to that of Park et al. (2020). Specifically, the nonlinear differential equations describing the heat-transfer dynamics are the following:

$$\begin{aligned}
\tau_h \dot{x}_1 &= a_1(T_\infty - x_1) + a_2(T_\infty^4 - x_1^4) + a_3(x_2 - x_1) + a_4(x_2^4 - x_1^4) + b_1 u_1 \\
\tau_h \dot{x}_2 &= a_1(T_\infty - x_2) + a_2(T_\infty^4 - x_2^4) + a_3(x_1 - x_2) + a_4(x_1^4 - x_2^4) + b_2 u_2 \\
\tau_c \dot{x}_3 &= x_1 - x_3 \\
\tau_c \dot{x}_4 &= x_2 - x_4
\end{aligned} \tag{16}$$

---

[5]Defined as the last 40% of data points of each trajectory.

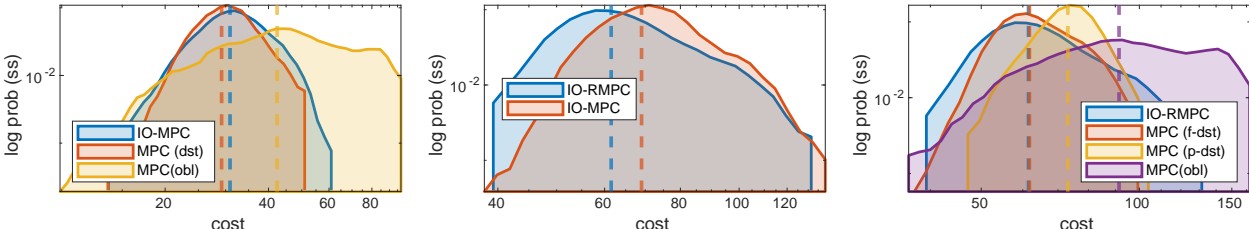

Figure 4: Steady-state cost distributions (log-log scale) over 100 trials of the experiments described in Section B.2. Dashed lines represent the median values. **Left:** MPC policies vs IO-MPC .**Center:** Difference in performance between the robust and non-robust version of IO policies when faced with distribution shift. **Right:** Performance of IO-RMPC vs MPC policies when faced with distribution shift.

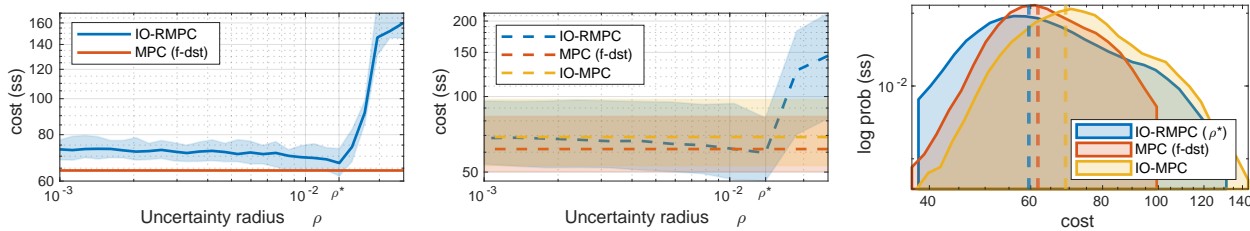

Figure 5: Additional experiments as described in Section B.2. **Left:** Time-averaged steady-state cost for different controllers trained with 50 different datasets and for varying $\varrho$; solid lines indicate the median values, and the tube indicates the range from the 5th to the 95th percentiles. **Center:** Steady-state cost distribution for different controllers trained with 1 dataset and for varying $\varrho$; the tubes consist of the 20th to the 80th percentile range from 100 trials, while the dashed lines represent the median values. **Right:** Steady-state cost histograms for optimal $\varrho = \varrho^*$ over 100 trials of a single controller realization; dashed lines indicate the median values.

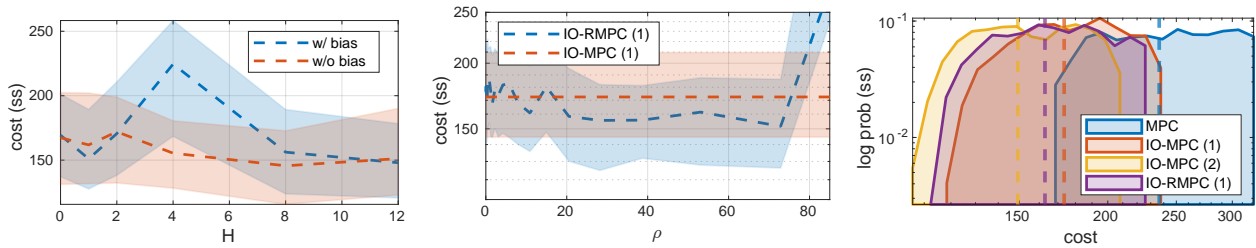

Figure 6: Experiments of Section B.3. **Left:** Steady-state cost distribution for different controllers trained with 1 dataset over 100 trials: we vary the size of $H$ and the effect the bias term has. **Center:** Steady-state cost distribution for different controllers trained with 1 dataset for varying $\varrho$ over 100 trials. **Right:** Steady-state cost histograms for the policies described in Section B.3 over 200 trials of a single controller realization. In all three figures, dashed lines indicate medians, and in the first two, the tubes consist of the range between the 20th and 80th percentiles.

| $a_1$ | $a_2$ | $a_3$ | $a_4$ | $b_1$ | $b_2$ | $\tau_c$ | $\tau_h$ |
|---|---|---|---|---|---|---|---|
| $4 \cdot 10^{-3}$ | $5.1 \cdot 10^{-11}$ | $7.3 \cdot 10^{-3}$ | $10^{-11}$ | $0.011$ | $0.006$ | $18.3$ | $2$ |

Table 7: Lumped-parameter coefficients of system (15).

with outputs $y_1 = x_3$ and $y_2 = x_4$. The parameters $a_1$, $a_2$, $a_3, a_4$, $b_1$, $b_2$, $\tau_c$, $\tau_h$, are lumped-parameter coefficients that can be summarized in Table 7. We assume full state feedback. The ambient temperature $T_\infty$ is constant throughout each trial, but randomly sampled from a uniform distribution $T_\infty \sim \mathcal{U}[18, 28]$, and is subjected to additional Gaussian noise $v_{t+1} \sim \mathcal{N}(0, 1)$ before entering the nonlinear dynamics. The control objective is for the outputs $y$ to track the temperature setpoints $r_1 = 55°C$ and $r_2 = 45°C$, with $Q_x = Q_f = I_2$ and $Q_u = \mathrm{diag}(1, 0.5)$. To obtain the nominal model $\tilde{f}_0$, we linearize (15) around $(\bar{x}, \bar{u})$ which corresponds to the steady-state solution of $y = r$, and then discretize with a sampling rate of $10\,\mathrm{s}$. As such, here the resulting nominal model $\tilde{f}_0$ used for the MPC controllers differs from the true nonlinear dynamics $f$. Due to this, the in-hindsight disturbance trajectories contain terms that stem from model mismatch:

$$w_{t+1} = E^\dagger \left( f(x_t, u_t, T_\infty + v_{t+1}) - \bar{x} - A\delta x_t - B\delta u_t \right)$$

where $\delta x_t = x_t - \bar{x}$, $\delta u_t = u_t - \bar{u}$ are its zero coordinates, on which our policies operate.

Similarly to before, we want to evaluate the performance of Inverse Optimization derived policies, in both the robust and non-robust settings. Specifically, we will investigate the performance of the following policies: **MPC**, a naive MPC with the assumption that $T_\infty = \mathbb{E}[T_\infty] = 23°C$; **IO-MPC (1)**, an IO-derived policy akin to (6) with feature map $\phi(\mathbf{x}_{1:t}, \mathbf{u}_{1:t}) = (\delta x_t, 1, \mathbf{w}_{t-1:t})$; **IO-MPC (2)**, like IO-MPC (1), but with no bias term and $H = 8$, thus $\phi(\mathbf{x}_{1:t}, \mathbf{u}_{1:t}) = (\delta x_t, \mathbf{w}_{t-7:t})$; **IO-RMPC (1)**, the robust counterpart to IO-MPC (1), equipped with $P = I_N$ and $\varrho = 70$. All IO-derived policies resulted from the same dataset, containing 10 trajectories of length 51 each.

Firstly, we performed an ablation on the features: whether or not to include a bias term and what is the best value of $H$ (lookback horizon). The results of this are present in the leftmost plot of Figure 6. It is evident that the optimal combination of features is no bias term and $H = 8$ (IO-MPC (2)). When evaluating the robust counterpart of IO-MPC (2), we found that for small values of $\varrho$, there was little to no performance improvement, and for larger values the performance deteriorated. We posit that given our experimental setting, IO-MPC (2) has enough expressivity that it can generalize well to unseen disturbances and capture most of the available performance, and thereby robustification has little benefit to add. On the other hand, when performing the same procedure on the worse-performing policy IO-MPC (1) with a bias term in the features and $H = 2$, we saw that robustification led to better generalization, as the performance improved when compared with its non-robust counterpart, as can be depicted in the middle plot of Figure 6.

Finally, in the rightmost plot of Figure 6, we can clearly see that each IO policy surpasses the performance of the naive approach (MPC), but that is to be expected as per our previous experimental discussions. The takeaway message from this figure is that robustifying can help in better generalization capabilities, and that our framework has the potential to deal with disturbance sequences that are correlated with the state, such as in cases where there is model mismatch.

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
