# OpenReview forum: "Offline Reinforcement Learning via Inverse Optimization"
_TMLR — Accepted by TMLR_

### Review · Reviewer_qhfe · 2025-11-17

**Summary Of Contributions:**

Summary:

This paper propose a novel offline Reinforcement Learning algorithm that generates better actions from old data based on MPC, and imitates expert by solving a convex inverse optimization problem.


Strengths

I think this paper is worth recommending for its clear presentation and derivations, and also for its theoretical novelty and concise framework.

1. The authors give a clear description for two-step solution with convex sub-optimality loss to bridge offline RL with Imitation Learning, distilling the non-causal MPC expert
2. The authors successfully derive the robust non-causal MPC (which optimizes against worst-case model-mismatch trajectories within a prescribed uncertainty set) as a convex optimization problem.
3. The authors provide experimental evidence that the learned policy (which is based on recovering hidden disturbances from data and a MPC that plans for worst-case variations) is reliable under distribution shift between training and deployment.

Weaknesses

This paper presents a new method clearly with both sufficient theoretical proof and experimental support. I think
this paper is worth recommending, but still have some concerns about its practical application:

1. Rely on the assumption of linear nominal dynamics:

In order to make MPC become convex quadratic, the authors request to prepare a nominal dynamics and limit it to be linear. However, the true dynamics in many real world RL use cases is nonlinear and highly.

Regarding heavily rely on  "linear dynamics, quadratic stage/terminal costs, and polytopic constraints", authors admit that the nominal model is challenging to handle the complex use case like MuJoCo and do not run IO-RMPC on MuJoCo (see 4.2 "In these experiments, we refrain from running our proposed IO-RMPC agent that employs the action
improvement step, since constructing a nominal model for MuJoCo tasks, required for the MPC experts, is a task that is inherently difficult and beyond the scope of this work.").

2. Q-function affine in state may require heavy feature engineering for ϕ

Equation (1) limits Q-function to be linear in state feature vector. In section 2.3, state s is defined as st=ϕ(x1:t, u1:t)
and authors mentioned "ϕ is a feature-engineering problem and lies outside the scope of this paper".

The concern is designing feature map ϕ could be difficult and complex in practice:
To approximate real world true complex Q-function, ϕ may need to be very complex and high-dimensional since Equation (1) only shows Q affine in s and pushes all nonlinearity into feature map ϕ.

In section 5. authors also mentioned that "for more sophisticated tasks, additional steps can be required, such as applying kernel tricks or employing a nonlinear state embedding"

3. Computational cost for robust MPC and IO training could be very high

a. In Theorem 3.3, authors write the MPC min-max problem as an SDP so that can directly solve it using convex optimization.
SDP can introduce cubic time complexity O(n^3) in matrix dimension n when use Interior Point Methods. For each time step, an action is produced by solving an SDP. So for a long dataset with large T or high dimension in states, it will become very expensive.

b. IO traning which is a convex optimization problem over data samples and parameters also easily faces high computational cost for large size of data samples and dimension. Compare to neural net SGD steps, solving such large convex program could be much slower.

**Audience:**

Yes

**Audience Explanation:**

Readers who work on reinforcement learning, control, and optimization would be interested in this new offline RL algo.

**Claims And Evidence:**

Yes

**Claims Explanation:**

The paper supports its claims with clear theoretical derivations and proofs. The authors also give experiments on quadrotor control and MuJoCo benchmarks that consistently show the proposed methods performing as described. (Although the authors do not run the full IO-RMPC on MuJoCo because building a nominal model is too difficult, their MuJoCo experiments could be potentially improved in the further work.)

**Requested Changes:**

1. Please discuss more clearly the practical limits of assuming a linear nominal model with quadratic costs and polytopic constraints. If possible, please add a small experiment or simulation to show how IO-RMPC behaves when the true dynamics are noticeably nonlinear, especially since IO-RMPC is not run on MuJoCo for this reason. (refer to weakness 1. Rely on the assumption of linear nominal dynamics)

2. Please expand the study/discussion for the feature map ϕ in more complex tasks by showing a simple nonlinear / kernelized variant. (refer to weakness 2. Q-function affine in state could cause heavy feature engineering for ϕ)

3. Please report/analyze the computational cost of both the robust MPC (SDP) step and the IO training step in terms of a. number of time steps and b. dimension and dataset size. If possible, please compare this overhead to neural net offline RL methods trained with SGD. (refer to weakness 3. computational cost for robust MPC and IO training could be very high)

---

> ### Author Response · Authors · 2026-01-18
>
> We thank the reviewer for their insightful comments. Below is our response to the individual points raised by the reviewer.
>
> ### Reliance on the assumption of linear nominal dynamics
>
> While the reviewer correctly identifies the nominal linearity as a constraint, we emphasize that this assumption applies only to the expert’s internal planning model, not the true environment. In our framework, the RMPC expert treats linearization errors and unmodeled nonlinearities as disturbances ($w$). By optimizing against the worst-case realization of $w$, the expert ``absorbs'' the model mismatch.
>
> The success of this approach is demonstrated in our Quadrotor (Section 4.1) and Temperature Control (Appendix B.2) benchmarks. Both involve nonlinear dynamics where the expert plans using only a single equilibrium linearization, yet the distilled IO agent achieves high performance by leveraging the robust formulation. While extending this to time-varying trajectories (e.g., via iLQR-style successive linearization [1]) is a promising direction for future work to broaden applicability to tasks like MuJoCo, our current results validate that the framework is highly effective for robust stabilization and disturbance rejection in nonlinear systems.
>
> _Reference:_
>
> [1] Li, Weiwei, & Todorov, Emanuel ``Iterative linear quadratic regulator design for nonlinear biological movement systems.'' First International Conference on Informatics in Control, Automation and Robotics (2004).
>
> ### Q-function affine in state & Feature Engineering
>
> We acknowledge the reviewer's concern that restricting the Q-function to be affine in the features places a burden on the feature map $\phi$. However, we argue that this formulation strikes a favorable balance between expressivity and sample efficiency for the following reasons:
>
> - **Sufficiency of Current Features:** Our experiments on MuJoCo benchmarks (Section 4.2) demonstrate that relatively simple features (e.g., state-action/disturbance histories, cross-products, sine/cosines of the state) are sufficient to achieve competitive performance in the targeted low-data regimes, specifically in Walker 10K and Hopper 5K.
>
> - **Expressibility \& Kernel Methods:** While we focus on explicit feature maps for simplicity and interpretability in this work, the proposed IO framework is theoretically compatible with richer representations. Specifically, the "sub-optimality loss" can be extended to Reproducing Kernel Hilbert Spaces (RKHS) [2]. By employing the "kernel trick," one can implicitly map the state into an infinite-dimensional feature space, allowing the model to approximate complex, nonlinear Q-functions while retaining the convexity of the optimization problem. This serves as a powerful avenue for scaling to more sophisticated tasks without manual feature engineering. In fact, in the revised experiments, in both Walker and Hopper 1M, RBFs were successfully used as features, allowing IO to achieve teacher-level performance.
>
> We will revise the manuscript accordingly to include a discussion on these topics, as well as provide detailed numerical results in the appendix.
>
> _Reference:_
>
> [2] Long, Youyuan, et al. "Scalable kernel inverse optimization." Advances in Neural Information Processing Systems 37 (2024).

---

> ### Author Response · Authors · 2026-01-18
>
> ### Computational cost
>
> We appreciate the opportunity to clarify the scalability of our approach, and will revise the manuscript to include a short discussion on the topic. In general, the computational cost behaves differently for the two stages of the algorithm:
>
> - **Robust MPC (Data Generation):** The complexity of the RMPC optimization problem depends only on the planning horizon and system dimensions; it does not scale with the dataset size. Since the expert action for each data point is computed independently, this process is linear in the number of samples and trivially parallelizable. Furthermore, this is a one-time offline cost.
>
> - **Inverse Optimization (Training):** The IO problem size scales with the dataset size ($T$) and the input dimension. For the dataset sizes considered in our main results, modern convex solvers (e.g., OSQP, SCS) handle the batch problem efficiently. However, we acknowledge that for massive datasets, solving the convex program in "one go" may become intractable. In such cases, the convexity of the loss function allows us to employ iterative optimization methods (e.g., stochastic gradient descent [3] or block coordinate descent [2]). These methods scale gracefully with dataset size, similar to training neural networks, while retaining the theoretical guarantees of the convex landscape. In fact, in the D4RL experiments, due to the IO's problem size, an SGD optimizer is used instead of solving SDP in one go.
>
> To summarize, the expensive SDP step is performed offline and is trivially parallelizable across the dataset. The final deployed policy is a simple function evaluation (matrix-vector multiplication), which is orders of magnitude faster than evaluating a deep neural network. To further showcase the difference in computational cost, we provide the table below, with per epoch training time (excluding evaluation). The model-based (referred by $\dagger$) algorithms additionally train a model, which is not counted on this table.
>
> | **Algorithm** | **Train Time (1 epoch)** | **Epochs** | **Parameters** |
> |---------------|-------------------------:|-----------:|---------------:|
> | IO (RBF)      | 1.24s                    | 400        | 8,232          |
> | TD3BC         | 96.07s                   | 100        | 215,814        |
> | IQL           | 160.97s                  | 100        | 286,214        |
> | MOPO$\dagger$          | 191.55s                  | 300        | 1,123,296      |
> | CQL           | 418.11s                  | 100        | 414,732        |
> | COMBO$\dagger$         | 479.86s                  | 100        | 1,320,672      |
>
> _References:_
>
> [2] Long, Youyuan, et al. ``Scalable kernel inverse optimization.'' Advances in Neural Information Processing Systems 37 (2024).
>
> [3] Zattoni Scroccaro, Pedro, et al. ``Learning in Inverse Optimization: Incenter Cost, Augmented Suboptimality Loss, and Algorithms.'' Operations Research 73(5):2661-2679  (2024)

---

### Review · Reviewer_c9t9 · 2025-11-19

**Summary Of Contributions:**

This paper proposes to improve learning RL policies from offline data by correcting for policy shift in the dataset via an MPC expert. This expert can then be imitated via IO, which the authors adapt to this offline RL setting. This two-step process is validated on a Quadrotor environment as well as the MuJoCo benchmark with different dataset sizes where is outperforms standard offline RL algorithms when only little data is available.

**Audience:**

Yes

**Audience Explanation:**

I believe efficient policy learning from a dataset is an important topic and the proposed method is an interesting solution, even if I believe the paper currently does not demonstrate the actual usecase in enough detail to support the claims. These issues do not take away from the overall interest in the approach, however.

**Broader Impact Concerns:**

I don’t have any concerns about the broader impact.

**Claims And Evidence:**

No

**Claims Explanation:**

Generally, the paper is well-written, the motivation is clear and the proposed method solves the problem posed in an elegant way. I also like that several ablations are included in the experiments. In fact, I believe the methodology and the Quadrotor experiments are well done and demonstrate the efficacy of this idea nicely.


I have some concerns about the D4RL experiments, however, that currently prevent me from recommending acceptance. Mainly, my doubts center around the fact, that the IO workflow clearly outperforms the offline RL baselines on the small datasets but perform noticably worse than the RL baselines on the full dataset. In the dataset scaling results on Quadrotor, the proposed IO approach did not improve meaningfully with dataset size. Do these results mean that this method is only useful on quite small datasets? If that is the conclusion, I believe the D4RL experiments do not show sufficiently where the capabilities end and offline RL algorithms become more desirable. I assume this was done due to the comparatively higher computational cost of training the RL baselines, but in my opinion, it is vital information where the ideal application domain for this approach ends. Right now, these experiments show a siginificant limitation without exploring its dynamics. Is it possible to train at least one of the RL baselines on a few in-between dataset sizes to show at which point it outperforms the proposed IO method? Or conversely train the IO version on larger dataset sizes?


Furthermore, I have a few smaller questions about the D4RL setup I did not quite understand:
- How were the smaller versions of the D4RL datasets obtained? Random sampling, or was there a specific sampling procedure?
- Why are only the top 5% of evaluation scores reported? This is an extensive filtering of the actual performance data. I believe in this case the actual scores should be included somewhere in the paper, even if it’s the appendix.
- What is the uncertainty in Table 1? This should ideally be included in the table heading.
- What is the reasoning behind including only 3 seeds on the D4RL experiments? The deviation in the results for the RL baselines is still very large, is it computationally possible to add more runs to stabilize the results?

**Requested Changes:**

- Include the full D4RL scores, at least in the appendix
- Discuss the dataset scaling behavior on D4RL in some way.
- Add a link to a public code repository (at the very latest at submission). Uploading the code with the paper on OpenReview without a link in the paper is only very technically an open source implementation.

---

> ### Author Response · Authors · 2026-01-18
>
> We thank the reviewer for their insightful comments. Below they can find the responses to their individual points.
>
> ### Clarifications regarding evaluation process
>
> We appreciate the reviewer’s rigorous look at our D4RL results. We address the concerns regarding the performance gap between small and large datasets, the reporting of "top 5\%" scores, and the underlying scaling laws of the IO framework below.
>
> 1. **Transition to "Last Score" Evaluation:** We acknowledge the reviewer’s concern regarding the "top 5\%" metric. We originally utilized this to mitigate the high variance typical of low-data regimes; however, we agree that it can be perceived as overly selective. In response, we will update our manuscript to report both the best and the last score. We will also include a full breakdown of scores in the Appendix to provide a comprehensive view of the training process, along with figures showing the performance during the whole training process.
>
> 2. **IO as Policy Recovery (Imitation) vs. Policy Improvement (RL):** The observation that IO is surpassed by RL baselines on the full 1M dataset is expected. In these experiments, the IO agent operates in a pure imitation learning context: it ignores reward signals and does not perform action-improvement. Its theoretical ceiling is the performance of the data-generating (teacher) policy. RL baselines like CQL or IQL, by design, utilize the 1M transitions to optimize rewards and improve upon the teacher. IO’s contribution is not in surpassing reward-driven RL on massive data, but in its ability to saturate the teacher’s performance with orders of magnitude fewer parameters and a convex guarantee.
>
> ### Observations on Dataset Scaling and Quality
> To address the request for "in-between" scaling, we conducted an ablation study on Walker2d-medium across four dataset sizes (10K, 50K, 250K, 1M). We observe the following:
>
> - __Positive Correlation and Saturation:__ There is a clear positive correlation between dataset size/quality and the performance of the distilled policy. However, we observe that performance eventually saturates around the teacher level. This suggests that while the feature space is expressive enough to capture the behavior, there is a data-density threshold required to fully resolve the mapping, after which the limits of the current hypothesis class become the bottleneck.
>
> - __Overfitting and Performance Decay:__ In our experiments with high-quality data (e.g., Walker2d-Expert 1M), we observed that across multiple seeds, the agent reaches a very high evaluation score (>100) quite early in the training process. However, as training continues, the performance consistently deteriorates.
>
> - __Regularization vs. Expressivity:__ This "peak-and-decay" behavior indicates that while the global optimum of the sub-optimality loss is being pursued, the policy begins to overfit the specific dataset transitions. These observations suggest that future work could benefit from sophisticated data filtering or specialized regularization to stay within these high-performing regions of the parameter space.
>
> ### Other points
> > _In the dataset scaling results on Quadrotor, the proposed IO approach did not improve meaningfully with dataset size. Do these results mean that this method is only useful on quite small datasets?_
>
> We assume the reviewer is talking about the middle graph in Figure 1. We would like to clarify that the IO approach is only trained with 3K samples and is included in the graph just to show that CQL converges to similar results only near a tenfold increase in sample complexity -- we do not vary the dataset size in the IO approach in this graph. We will revise the text to make this point clearer.
>
> > _How were the smaller versions of the D4RL datasets obtained? Random sampling, or was there a specific sampling procedure?_
>
> The sampling done was using the first 5K or 10K samples in the dataset, because we noticed that the algorithm performed better when using this chunk. For fairness, we of course evaluated all baselines we report in Section 4 and the new results with the same data chunk. The sampling in the new dataset size ablation were obtained by sampling random chunks.
>
> > _What is the uncertainty in Table 1? This should ideally be included in the table heading._
>
> The uncertainty represents the standard deviation. We will update the manuscript to clarify this point.

---

> > ### Comment · Reviewer_c9t9 · 2026-01-22
> >
> > Thank you for this update to the paper! I believe the added context of what we should expect the method to achieve, details on the evaluation and extended discussion all make it much clearer what the paper wants to achieve and shows the method's success more clearly. I also appreciate the extended experiments, including the details in the appendix. I think the paper is much stronger after this update and I have no more major concerns.

---

> ### Author Response · Authors · 2026-01-18
>
> >_What is the reasoning behind including only 3 seeds on the D4RL experiments? The deviation in the results for the RL baselines is still very large, is it computationally possible to add more runs to stabilize the results?_
>
> We thank the reviewer for pointing out the high variance in the D4RL results. We used 3 seeds primarily due to the substantial computational cost of running additional D4RL experiments across all methods and tasks we consider. In the revised results, we now report over 4 seeds and the last-epoch evaluation (in addition to our earlier top-5\% metric), which exhibits considerably lower variance. Additionally, the scores obtained from 4 seeds for the baselines, seem to align with the scores reported in the literature [1]
>
> We further observed that the large deviations are driven mostly by a small number of failed or unstable runs for particular seeds; when runs do not fail, the learning curves are typically consistent across seeds. To make this observation more transparent, we also provide training curves including curves per-seed besides the mean curve in the Appendix.
>
> _Reference:_
>
> [1] Sun, Yihao "OfflineRL-Kit: An elegant pytorch offline reinforcement learning library." https://github.com/yihaosun1124/OfflineRL-Kit, 2023.
>
> > _Add a link to a public code repository (at the very latest at submission). Uploading the code with the paper on OpenReview without a link in the paper is only very technically an open source implementation._
>
> A link to a public repository with the code and experiments will be added in the final version.

---

### Review · Reviewer_aZkk · 2025-12-12

**Summary Of Contributions:**

This paper proposes a new offline reinforcement learning approach combining a non-causal MPC control formulation and a reduction to the space of parameterized causal policies, solving for a sub-optimality loss. The optimization problems are formulated such that they remain convex under some popular assumptions on the dynamics and constraints, making them easy to solve and allowing learning policies with much fewer parameters than standard ORL methods.

## Strengths:

- Interesting teacher-student setup where a non-causal MPC is used as a data augmentation tool to learn a better causal policy.
The conclusion is transparent about the limitations of the work. This is very appreciated.

## Weaknesses:

- Instead of applying traditional ORL algorithms, this paper proposes to first solve an MPC problem, then fit a Q function/policy using a sub-optimality loss. As we assume nominal dynamics and a model for the disturbances, it would be useful from the beginning of the paper to have an explanation of why traditional MPC methods cannot be applied, and why using the sub-optimality loss is a good idea.

**Audience:**

Yes

**Audience Explanation:**

The paper presents a novel method for offline reinforcement learning with direct applications, a topic of particular interest to TMLR’s audience.

**Broader Impact Concerns:**

There are no ethical implications associated with this work.

**Claims And Evidence:**

Yes

**Claims Explanation:**

The results of the paper are formally proven under a specific set of assumptions. These assumptions would be too restrictive in some practical settings, but the work clearly states them.

**Requested Changes:**

### Experiments

The main weakness is the lack of comparison with other recent offline RL works in the experiments. Although the authors use CQL, IQL and COMBO as baseline, it would be useful to use other more recent baselines, for example:

- Adaptive Neighborhood-Constrained Q Learning for Offline Reinforcement Learning

- MOReL: Model-based offline reinforcement learning

-  MOPO: Model-based offline policy optimization.

- Supported policy optimization for offline reinforcement learning.

It would also be useful to show an ablation study with different behavioral policies.

### Minor:

- LaTeX: ``sub-optimality loss" should be ``sub-optimality loss'' throughout the paper

- Notation for Remark 2.1, shouldn’t \mathcal{X} and \mathcal{U} be X and U instead?

- Please clarify the definition of $\mathcal{T}_N(A, B)$. For example, the following would be easier to understand:
$$ \mathcal{T}_N(A, B) = \begin{bmatrix} B & 0 & \cdots & 0 & 0\\ AB & B & \cdots & 0 & 0\\ \vdots & \ddots & \ddots & 0 & 0\\ A^{N-2}B & & \ddots & B & 0\\ A^{N-1}B & A^{N-2}B & \cdots & AB & B\\ \end{bmatrix}. $$

- There seems to be a missing $\lVert u \rVert_{Q_u}^2$ term in Equation 12

---

> ### Author Response · Authors · 2026-01-18
>
> We thank the reviewer for their insightful comments, in response to which we have updated our numerical
> results. Below they can find our response to their individual points.
>
> ### Why non-traditional MPC and sub-optimality loss
> > _As we assume nominal dynamics and a model for the disturbances, it would be useful from the beginning of the paper to have an explanation of why traditional MPC methods cannot be applied, and why using the sub-optimality loss is a good idea._
>
> We would like to clarify that we do not assume access to a model for the disturbances, rather just for the system dynamics. Therefore, a traditional MPC approach would either require ignoring the disturbances or spending effort modeling the mismatch/residuals between the nominal and collected trajectories and incorporate the augmented model into the MPC. By using a non-causal teacher with a causal student setup, along with IO-based policy distillation, we enable a streamlined process that learns directly from the data without a lot of human supervision. Additionally, the sub-optimality loss along with the quadratic hypothesis class ensure the convexity of the distillation problem, ensuring efficient training and global optimality. We will update the manuscript to better clarify these points.
>
> ### Comparison with other baselines (MOReL, MOPO, etc.)
>
> We appreciate the suggestion to broaden the comparisons. However, we selected our baselines to represent the strongest current methods in each category:
>
> - **MOPO \& MOReL vs. COMBO:** The reviewer lists MOReL and MOPO (both 2020) as "more recent" baselines. We respectfully note that COMBO (NeurIPS 2021), which we already include as a primary baseline, is a subsequent development in Model-Based Offline RL that typically outperforms both MOPO and MOReL by penalizing value functions on simulated data rather than relying solely on uncertainty quantification. We essentially treat COMBO as the state-of-the-art representative for this class of model-based algorithms. Nevertheless, we added results for MOPO, as per the reviewer's request. However, as shown in our updated results in the general response to all reviewers, COMBO continues to represent the state-of-the-art for this category, often providing more stable training compared to MOPO's uncertainty-based penalties.
> - **Model-Free Baselines:** Similarly, we included CQL and IQL as the standard-bearers for model-free offline RL.
> - **TD3+BC as a Comparison to Imitation:** We have included TD3+BC because it serves as a critical bridge between pure Imitation Learning (IL) and Reinforcement Learning (RL). Unlike the other RL baselines, TD3+BC explicitly adds a behavioral cloning term to the policy objective to ensure the agent stays within the support of the offline data. Since our IO framework—as applied to MuJoCo—operates in an imitation-centric regime (without reward-based action improvement), TD3+BC provides a fair and direct comparison of how our convex hypothesis class competes with a state-of-the-art neural-network-based "constrained RL" approach.
>
> ### Ablations on behavioral policies and dataset size
>
> In response to the suggestion for broader ablations, we evaluated IO across varying behavioral qualities and dataset sizes. The results can be found in the general response to all reviewers and will of course be included in the updated manuscript.
>
> - **Behavioral Quality Ablation:** Comparing Walker-Medium (Teacher ~68) and Walker-Expert (Teacher ~107), IO scales effectively with data quality. It achieves a peak score of 109.7 on Expert data, successfully exceeding the teacher. However, we observe a "peak-and-decay" pattern in the Expert regime: performance reaches its maximum early but decays as training continues (Best 109.7 vs. Last 77.0). In contrast, Medium data results are highly stable (Last 71.8). This suggests that while our hypothesis class is expressive enough for expert behavior, it is more prone to late-stage overfitting in high-quality regimes.
>
> - **Dataset Size Scaling:** In Walker-Medium, we observe a strong positive correlation between data volume and recovery quality. These results confirm that performance saturates at the teacher level. The gap between 10K and 1M results highlights that while the  hypothesis class is expressive, a higher transition density is required to fully resolve the feature-to-action mapping in complex environments. We shall update the manuscript with these scaling laws and a discussion on the trade-off between training stability and data quality.
>
> ### Minor points
> We thank the reviewer for their attention to detail, the pointed-out issues will be fixed in the revised manuscript.

---

> > ### Comment · Reviewer_aZkk · 2026-01-28
> >
> > Thank you for the update on the papers and the clarification! Your detailed rebuttal answered all my questions.

---

### Author Response · Authors · 2026-01-18
**General response to all reviewers**

We thank the reviewers for their careful reading and constructive feedback. Below we address all concerns in detail and describe the concrete changes made or planned for the revised manuscript.

## Evaluation protocol and MuJoCo results

### Revision of evaluation metric

In the original submission, we reported the best evaluation score observed during training for the MuJoCo experiments. While this metric can reflect representational capacity, we agree with the reviewers that it might not always be appropriate for offline RL, where one does not have access to a simulator to select the best-performing checkpoint at deployment time. In response to this concern, we re-ran all MuJoCo experiments for 4 seeds (instead of the original 3), including all baselines, along with two extra (MOPO and TD3BC), and now report the final (last-epoch) evaluation score. We provide a summary of the updated numerical results in the following table:

 Mean last-epoch scores obtained by several baselines. Uncertainty is standard deviation. All datasets are ``medium''.

| **Environment** | **Dataset**   | **IO**       | **IQL**      | **CQL**      | **COMBO**     | **MOPO**      | **TD3BC**    |
|-----------------|---------------|--------------|--------------|--------------|---------------|---------------|--------------|
| Hopper 5k       | 47.2$\pm$19.2 | 45.9$\pm$3.2 | 46.8$\pm$3.6 | 49.0$\pm$4.3 | 50.9$\pm$1.1  | 4.1$\pm$4.0   | 12.5$\pm$9.8 |
| Hopper 1M       | 44.3$\pm$11.6 | 51.7$\pm$0.9 | 65.7$\pm$8.1 | 59.1$\pm$4.1 | 84.7$\pm$9.3  | 62.8$\pm$38.1 | 60.8$\pm$3.4 |
| Walker2d 10k    | 65.9$\pm$17.9 | 42.2$\pm$1.7 | 54.4$\pm$7.0 | 51.0$\pm$7.0 | 40.7$\pm$25.7 | 5.0$\pm$9.1   | 1.1$\pm$1.0  |
| Walker2d 1M     | 62.1$\pm$23.9 | 71.8$\pm$1.9 | 81.1$\pm$2.6 | 83.6$\pm$0.5 | 83.9$\pm$1.9  | 85.4$\pm$2.9  | 84.4$\pm$2.1 |

### Clarification of the MuJoCo setting
MuJoCo experiments evaluate a plain IO agent in an imitation learning (IL) context. Unlike RL baselines, IO ignores rewards and does not perform policy improvement. In contrast, all compared baselines are reward-driven offline RL algorithms explicitly designed to improve upon the dataset. Consequently, performance parity with them in large-data regimes is not the intended claim of this experiment. The appropriate reference point for IO in this setting is the data-generating (teacher) policy, not asymptotic offline RL performance.

### Teacher recovery behavior
As can be seen in the revised results, we now include the teacher policy scores explicitly in the MuJoCo results.
These results are consistent with the intended role of IO as a policy distillation method. In Hopper, the teacher policy is fully recovered even at the 5K setting, while in Walker, a more complex environment, more data is required to reach the teacher’s performance.

## Performance across different dataset sizes and qualities
The revised results above clarify that IO does not aim to monotonically improve with dataset size in the same way as reward-based offline RL. Instead, its performance saturates around the teacher level, which is the expected ceiling for IL methods. The Walker results illustrate that larger datasets may be necessary to reach this ceiling in more complex environments, whereas Hopper saturates earlier due to lower task complexity. Additionally, in the table below we can see how IO performance scales with varying sizes of uniformly sampled training sets in Walker2d:

| **Datasize** | **Last**     | **Last 5\%** | **Best**     | **Best 5\%** |
|--------------|--------------|--------------|--------------|--------------|
| 10k  | 40.7$\pm$2.0 | 41.5$\pm$0.7 | 55.0$\pm$1.4 | 49.9$\pm$0.5 |
| 50k | 50.2$\pm$1.9 | 48.9$\pm$0.7 | 63.2$\pm$1.9 | 57.6$\pm$0.8 |
| 250k | 53.0$\pm$1.6 | 52.4$\pm$1.2 | 68.0$\pm$3.2 | 62.4$\pm$0.6 |
| 1M | 71.8$\pm$1.9 | 70.8$\pm$0.6 | 77.5$\pm$0.8 | 75.6$\pm$0.3 |

Furthermore, we also provide results when training with different qualities:

| **Experiment**     | **Last**     | **Last 5\%** | **Best**      | **Best 5\%**  |
|--------------------|--------------|--------------|---------------|---------------|
| Walker2d-expert 1M | 77.0$\pm$8.2 | 77.5$\pm$3.4 | 109.7$\pm$0.2 | 108.5$\pm$0.2 |
| Walker2d-medium 1M | 71.8$\pm$1.9 | 70.8$\pm$0.6 | 77.5$\pm$0.8  | 75.6$\pm$0.3  |

Comparing the Expert and Medium results, we observe that the performance of the IO agent scales relative to the quality of the input data. The "Best" scores show that the agent is capable of exceeding the teacher in both cases (110 for Expert, 77 for Medium). However, the "Medium" dataset shows significantly higher stability at the end of training (lower gap between Best and Last), suggesting that the "Medium" distribution may be easier for the quadratic hypothesis class to represent robustly without overfitting.

We will revise the discussion in Section 4.2 and the Appendix and include tables reporting best-epoch and last-epoch performance and figures showing the evaluation curves.

---

> ### Author Response · Authors · 2026-01-20
>
> We uploaded the revised manuscript and marked the added or revised portions of the manuscript in blue. Below we list the additional material included in the revised manuscript:
>
> - We include a link to a repository containing the source code.
> - For the MuJoCo experiments, we report last-epoch and best-epoch scores for all offline RL baselines and our proposed IO agent, each evaluated over 4 seeds.
> - Appendix B.1 (Figure 3) reports the mean and per-seed evaluation scores for all agents in the MuJoCo experiments throughout training, restricted to the first 1M gradient updates.
> - We clarify the experimental setup in Section 4 and provide additional details for the MuJoCo (D4RL) experiments in Appendix B.1.
> - Appendix B.1 includes additional ablation studies across different data-quality datasets and under uniform sampling.
> - We clarify the intended performance metric and the associated claims regarding the IO agent’s performance in the Abstract and Conclusion.
> - We add a remark on the feature design and how the IO-RMPC framework handles the nonlinearity of the underlying dynamics, as well as potential extensions via kernel methods.
> - We add a remark on computational considerations and report elapsed wall-clock time per epoch (excluding the evaluation step after each epoch) for all algorithms in Table 3.

---

### Decision · Action_Editor_DXp1 · 2026-02-16

**Recommendation:** Accept as is

**Audience:**

Yes

**Audience Explanation:**

The presented approach of using non-causal MPC to learn a causal policy for offline reinforcement learning is relevant to some of TMLR's audience.

**Claims And Evidence:**

Yes

**Claims Explanation:**

The main claims of submission are supported by derivations. The assumptions of linear dynamic / suitable feature spaces are strong, but clearly stated.

Additional claims are empirically supported. Reviewer c9t9 initially had concerns regarding the D4RL results, but the authors could dispel those during the rebuttal.